# Transcriptional Regulation of *CCL2* by PARP1 Is a Driver for Invasiveness in Breast Cancer

**DOI:** 10.3390/cancers12051317

**Published:** 2020-05-21

**Authors:** Pranabananda Dutta, Kimberly Paico, Gabriela Gomez, Yanyuan Wu, Jaydutt V. Vadgama

**Affiliations:** 1Division of Cancer Research and Training, Department of Medicine, Charles R. Drew University of Medicine and Science, Los Angeles, CA 90059, USA; pranabandutta@cdrewu.edu (P.D.); kpaico@ucmerced.edu (K.P.); gabrielagomez@cdrewu.edu (G.G.); yanyuanwu@cdrewu.edu (Y.W.); 2Jonsson Comprehensive Cancer Center, David Geffen School of Medicine, the University of California at Los Angeles, Los Angeles, CA 90059, USA

**Keywords:** PARP1, CCL2, TNBC, metastasis, breast cancer, invasiveness

## Abstract

Background: PolyADP ribosylation (PARylation) by PARP1 is a significant post-translational modification affecting protein function in various cancers. However, PARP1 mediated cellular processes in the context of breast cancer are not fully understood. Method: To identify potential targets of PARP1, we carried out whole transcriptome sequencing with shRNA mediated PARP1 knockdown in triple-negative breast cancer (TNBC) cell line and inhibited PARP1 with a known PARP1 inhibitor, PJ34. Results: Analysis of the transcriptomics data revealed that PARP1 is involved in regulating multiple chemokines under basal conditions, including the chemokine ligand 2 (*CCL2*). PARP1 knockdown and PJ34 mediated inhibition showed reduced *CCL2* transcript levels in breast cancer cells, corroborating the findings from the sequencing data. We further showed that PARP1 interacts with the NFκB P65 subunit to regulate transcription of *CCL2*. Using chromatin immunoprecipitation, we confirm that both PARP1 and P65 localize to the promoter of *CCL2*, suggesting direct regulation of *CCL2* promoter activity. CCL2, in turn, can positively affect the PARP1 pathway, as global PARylation levels increased upon CCL2 treatment. Conclusion: Our results indicate crosstalk between PARP1 and CCL2, which is critical for maintaining CCL2 levels in breast cancer cells and subsequently drives cellular invasiveness.

## 1. Introduction

Multiple studies have shown the involvement of the small secreted proteins like cytokines and associated receptors to be involved in cell proliferation and metastasis, causing poor prognosis [1]. In breast cancer, along with their direct effects on cell growth and metastasis, chemokines are also essential mediators in the tumor microenvironment, maintaining crosstalk in the tumor niche [2]. Aggressive types of breast cancer, like TNBCs, have been shown to have “secretome” comprised of higher concentrations of chemokines such as CXCL1 and CCL2 [3,4]. Understanding how these chemokines are maintained and upregulated in breast cancer cells can generate some strategies for therapies. 

Poly ADP ribose Polymerase 1 (PARP1) belongs to a large family of proteins that have 17 family members in humans. PARP1 mediated ADP ribosylation is a critical step in single-strand D.N.A. damage repair, and its inhibition is synthetically lethal in BRCA1 deficient breast and ovarian cancers [5]. Higher nuclear vs. cytoplasmic PARP1 levels appear to be unfavorable in lymph node-negative breast cancers [6]. Recent evidence indicates that PARP1 is a functional meditator of cell signaling and transcriptional mechanisms, including chromatin regulation, cell survival, movement, and metastasis processes. *PARP1* expression or protein level is relevant for tumor prognosis. In basal conditions, PARP1 also regulates transcriptional activity in cancer cells [7]. For example, PARP1 is known to be downstream of ER-dependent transcriptional response in breast cancer cells [8].

Interestingly, PARP1 controls inflammatory cytokine transcription during senescence along with NFκB in melanoma cells. A vital component of this senescence-associated secretory phenotype (SASP) is the chemokine CCL2 [9]. CCL2 is a small 17kd secreted protein that acts via G-protein coupled receptor CCR2 for downstream signaling. Importantly, CCL2, along with other inflammatory cytokines, is a modulator of cancer invasiveness by affecting tumor microenvironment, and its higher expression predicts worse outcomes for breast cancer patients. 

CCL2 is also known to be a contributing factor promoting epithelial-mesenchymal transition and metastatic potential in triple-negative breast cancer (TNBC) [3,4]. TNBCs lack any targeted therapy due to lack of receptor expression and also contribute to health disparity as African-American women are at a higher risk of developing this type of breast cancer. However, how *CCL2* expression increases in breast cancer, particularly in TNBC, is not fully understood. Here we show that PARP1 is an essential mediator of *CCL2* transcription. Our data show that PARP1 and transcription factor NFκB P65 subunit regulate *CCL2* transcription activity. We further provide evidence that CCL2 can affect PARP1 function, possibly via MAP kinase (ERK1/2) signaling. Thus, our work indicates therapeutic inhibition of PARP1 in patients with upregulated *CCL2* might be useful in reducing metastasis, thereby lowering the risk of disease recurrence.

## 2. Results

### 2.1. PARP1 Inhibition Negatively Affect Breast Cancer Cell Proliferation and Migration

We examined the total levels of PAR and PARP1 in cell lysates from different subtypes of breast cancer cells. Interestingly, PAR levels were higher in triple-negative breast cancer cells, as shown on the western blot (Figure 1A). To account for the differences in PARylated proteins, we also examined total PARP1 levels in the cells. However, the levels of PARP1 were not higher in TNBC cells. Next, we investigated whether the PARP1 function is essential for breast cancer cells. To this end, we performed cell proliferation assay at 48 h, and 72 h intervals with MDA-MB-231 (MB-231) cells treated with PJ34 PARP1 inhibitor [10] (Figure 1B). Figure 1B shows the non-linear regression curve for PJ34 mediated inhibition. MB-231 cells were treated with various doses starting from 6.5 µM to 50 µM. We observed dose-dependent growth inhibition in MB-231 cells with an IC50 value of ~27 µM for 72 h treatment as determined by four parametric regression lines (Figure 1B). This could be attributed to cell proliferation defect, as overnight treatment with PJ34 did not induce any significant apoptosis (Appendix A). In the low attachment plates, long term (7 days) treatment with 25 µM PJ34 also resulted in a smaller number of colonies compared to untreated vehicle control (hereafter “untreated”) cells seeded at 1000 cell/well density (Figure 1B right panel). Next, we investigated the effect of PARP1 inhibition on cell migration. To this end, MB-231 cells pretreated with PJ34 were also subjected to migration assay (Figure 1C, left) and invasion Assay (Figure 1C right). PJ34 treated MB-231 cells failed to migrate as fast as untreated cells in wound healing assay as seen by higher wound width (White dotted line) after 10 h, post wound creation (Figure 1C). Pre-treatment with PJ34 at 20 µM doses significantly reduced cell invasion in the Boyden chamber assay with mean invading cell numbers reduced to 4 from 17 when treated (Figure 1C right graph). 

To understand the cellular pathways affected by PARP1 activity, we performed whole transcriptome sequencing utilizing shRNA against PARP1 in TNBC BT549 cells. Along with that, we also used RNA harvested from MB-231 cells with PJ34 at 10 µM doses overnight (Appendix A). Isolated RNA was subjected to whole transcriptome sequencing (Appendix A: Gene lists for the whole transcriptome analysis.). PARP1 knockdown showed a significant effect on cytokine signaling. In the Ingenuity Pathway Analysis (IPA), interferon signaling and IL-17A were significantly affected, showing 25% and 23% overlap with the genes in those pathways, respectively (Figure 1D). Interferon pathways showed predicted activation with IL-17A did no present any pattern in IPA analysis. *PARP1* knockdown similarly affected cellular processes involving antigen presentation and activation of IRF (Interferon Regulatory Factors). In these cases, the overall change in the pathways remained neutral due to a lack of published literature. Overall the data suggested that PARP1 can modulate inflammation-related pathways. More work will be needed to understand the mechanisms involved.

On the other hand, IPA showed that the significant pathways upregulated in MB-231 are the EIF2 signaling with 42% overlap, oxidative phosphorylation showing 49% overlap with the genes of those pathways (Appendix A). Protein translation-related EIF4 and p70S6K pathways were also positively regulated. Along with that, Sirtuin signaling showed significant changes. This might be due to PARP1 consuming NAD+, an essential component of the Sirtuin pathways [11].

### 2.2. PARP1 Inhibition Affects Cytokine-Mediated Cellular Signaling in Breast Cancer Cells

Next, we explored the overlap between PJ34 mediated gene signatures and differentially expressed genes in response to *PARP1* knockdown. A total of 1095 genes were found to be differentially expressed in BT549 cells upon *PARP1* knockdown, as determined by the edgeR method [12]. On the other hand, PJ34 treatment in MB-231 cells significantly affected the transcriptional status of nearly 5759 genes. However, we found that 656 genes are expressed in both of these differentially expressed gene sets (Appendix A). Thus, this set of overlapping genes could be a direct target of PARP1 mediated regulation.

We subjected these common genes to Gene Set Enrichment Analysis (GSEA), primarily using the Broad Institute Oncogenic Signature set of genes [13]. Interestingly, genes that were differentially affected by either PJ34 treatment or *PARP1* knockdown showed significant enrichment for VEGF_A_UP.V1_UP, TGFB_UP.V1_DN gene set as well as IL2_UP.V1_UP, among others. These genes are putatively associated with the inflammatory signature. Most of these genes were differentially expressed upon treatment with cytokines like *VEGF*, *TGFβ*, and *IL2* (Appendix A). Leading-edge analysis (for genes that contributed most to enrichment in pathways) showed genes that are upregulated downstream of KRAS in lung or breast cancer (e.g., *CXCL3*, *ADAM8*, *and L.I.F.*) (Appendix A). Genes in TGFβ and polycomb repressive complex (P.R.C.) pathways were enriched in the PJ34-PARP1 shRNA overlapping set as well.

In our transcriptomic analysis, we also found that multiple cytokines are differentially expressed with PARP1 inhibition and knockdown (Figure 1D bottom left). A specific pattern of gene expression was not observed among these cytokines as both upregulation and downregulation were observed in the transcriptome analysis. For example, *CCL2* and *CCL3* were downregulated in all conditions (Figure 1D bottom panel). However, various signaling pathways were affected, including cell death, survival, cellular movement, antigen presentation (Figure 1D bottom right). Interestingly, *CCL2* is one of the cytokines that showed downregulation with both PJ34 treatment and PARP1 knockdown. Hence we speculated that PARP1 is a transcriptional regulator of *CCL2* in breast cancer cells and focused on interpreting the underlying mechanisms of *CCL2* regulation.

### 2.3. PARP1 Transcriptionally Regulates CCL2 in Breast Cancer

To examine the role of PARP1 in *CCL2* transcription, we utilized siRNAs to knockdown *PARP1* levels in MB-231 and BT549 cells (Figure 2A). PARP1 protein levels were significantly reduced upon siRNA treatment in cells at 72hr after transfection using two different siRNAs targeting PARP1 mRNA. Importantly, the total PAR level (an indicator of PARylated proteins) was also reduced, indicating the loss of PARP1 activity (Figure 2B). We, therefore, isolated RNA from these siRNA transfected cells and subjected these to quantitative real-time polymerase chain reaction (RTPCR) for *CCL2* transcript levels. Interestingly, siRNA mediated *PARP1* knockdown reduced *CCL2* transcript significantly in both cell lines (Figure 2C,D). We found an overall 90% decrease in *CCL2* concerning scrambled control in MB-231 cells and at least 80% downregulation in BT549 cells upon siRNA transfection. Our result corroborates the findings of transcriptome analysis and shows the involvement of PARP1 in *CCL2* transcription.

### 2.4. PAR Level Is Associated with CCL2 Transcription in Breast Cancer

Next, we asked whether PARP1 enzymatic activity is associated with *CCL2* mRNA levels. To this end, we used PJ34 to inhibit PARP1. We examined the PAR level as a surrogate for PARP1 activity. As seen in Figure 3A, PJ34 treatment resulted in a reduction in total PAR levels in MB-231 cells in western blots (Left panel). Concurrently, a decrease in *CCL2* transcript level was observed in a dose-dependent manner when treated with 5 and 10 µM PJ34 (Figure 3A Right Panel). We also used enzyme-Linked Immunosorbent Assay (ELISA) to measure secreted CCL2 protein in cell-conditioned media. With 10 µM PJ34 treatment overnight, the total amount CCL2 in the cell-conditioned media significantly decreased from 4094 pg/mL to 1562 pg/mL (Figure 3B). Thus, inhibiting PARP1 resulted in lower CCL2 levels. We also observed the downregulation of *CCL2* mRNA in BRCA1 mutated TNBC cell lines HCC1937 and MDA-MB-436 cells with PJ34 treatment (Appendix A). While *CCL2* levels were decreased in both cell lines, we found an 80% to 90% reduction in *CCL2* in the HCC1937 cells. In the HCC1937 cells, PARP1 was also localized to the *CCL2* promoter as seen by chromatin immunoprecipitation with PARP1 antibody (Appendix A). We then asked whether increasing overall PAR levels can affect *CCL2* levels. To this end, we utilized Poly-ADP-ribose glycohydrolase (PARG) inhibitors to treat MB-231 cells. Since PARG is a hydrolase that de-PARylates proteins by removing PAR moiety, we hypothesized that PARG inhibition would increase *CCL2* transcript [14]. Interestingly, treating MB-231 cells with PDD00017273 (PARGi) resulted in increased PARylation (Figure 3C left) [15]. Interestingly, *CCL2* transcript levels increased with both 5 µM and 10 µM PARGi doses (~50% increase, *p* < 0.05) along with an increase in PARylation (Figure 3C right panel). Next, we asked whether PARG treatment can prevent PJ34 mediated downregulation of the CCL2 mRNA. To this end, we subjected MB-231 cells to treatment with both PJ34 and PARGi. Cells were pretreated with 10 µM PJ34 overnight, followed by PARGi at a 5 µM dose. Interestingly, PARGi and PJ34 treatment resulted in an increased expression of *CCL2* compared to PJ34 alone (Figure 3D). Thus, PARG inhibition partially rescued PJ34 mediated downregulation of *CCL2*, suggesting a positive correlation between *CCL2* transcription and PARylation in the breast cancer cells.

### 2.5. PARP1 and NFκB Interaction is Essential for CCL2 Transcription in Breast Cancer

The canonical NFκB signaling, particularly involving the Rel-A subunit (hereafter P65), is necessary for maintaining inflammatory cytokine transcription [16]. In melanoma, PARP1 is mainly known to regulate NFκB activity, thus affecting *CXCL1* cytokine transcription [17]. Evidence also indicates that PARP1 and NFκB cooperate during senescence to increase the secretion of multiple cytokines, including CCL2 [9]. We found that the *CCL2* promoter has an NFκB binding motif (GGGAGGCCCC) at 304 to 310 bp upstream of the transcription start site. We first treated MB-231 cells with Bay 11-7082 (NFκBi) small-molecule inhibitor of the P65 subunit [18]. NFκBi treatment resulted in the downregulation of *CCL2* transcripts with 2.5 µM and 5 µM doses showing 60% and 50% reduction compared to untreated controls (Figure 4A), indicating the involvement of NFκB pathway. We then asked whether NFκB is affected upon PARP1 inhibition. To this end, we treated MB-231 cells with PJ34 and with 50 ng/mL recombinant TNFα (to activate the NFκB pathway).

Interestingly, in the presence of the PARP1 inhibitor, P65 serine 536 phosphorylation, associated with NFκB activation, was reduced significantly without affecting the total level of P65 (Figure 4B left panel). Thus, it appears that PARP1 inhibition resulted in the attenuation of NFκB signaling. We then asked whether PARP inhibition will also attenuate P65 transcriptional activity. To this end, we employed an Electrophoretic mobility shift assay (EMSA) with biotinylated probes for P65. We observed that there is a significant reduction in P65 bound probe (arrow Figure 4B Right panel) when treated with PJ34. PARP1 is known to modulate the NFκB transcription factor by PARylation [17] and also by binding to the p65 subunit [19]. Therefore, we wanted to test whether, in breast cancer p65 interacts with PARP1. To this end, we treated MB-231 cells with PJ34 and isolated cytosolic and nuclear fraction after inhibiting nuclear export with leptomycin B (100 ng/mL) for 1 h. We subjected nuclear lysate to immunoprecipitation with the P65 antibody (Figure 4C). P65 and PARP1 interactions were observed in untreated samples, shown in Figure 4C. Overnight PJ34 treatment led to a reduction in the PARP1 and p65 interaction. Thus PARP1 inhibition negatively affects P65 PARP1 interaction and results in loss of P65 DNA binding activity. Since we observed that P65 and PARP1 interact in breast cancer cells, we wanted to examine whether it is bound at the CCL2 promoter and if the binding is affected due to PARP1 inhibition. To determine this, we stimulated MB-231 cells with TNFα with or without overnight PJ34 pre-treatment. As expected, we observed increased binding of P65 after 1 hr of TNFα treatment by chromatin immunoprecipitation (IP) (Figure 4D left). However, PJ34 pre-treatment significantly reduced TNFα dependent P65 binding at the promoter.

Along with P65, PARP1 also localized to the same locus as observed by chromatin IP with PARP1 specific antibody (Figure 4D Right), which decreased upon PJ34 treatment. This observation is also consistent with globally reduced chromatin occupancy of PARP1 upon PJ34 treatment. Cytosolic and nuclear fractions from MB-231 cell lysate treated with PJ34 showed a reduction in the nuclear fraction of PARP1 and P65 (Appendix A, whole western blot in Appendix A). Thus, PARP1 might also be affecting *CCL2* transcription by modulating nuclear transport of P65 and its transcription factor activity.

### 2.6. Crosstalk between PARP1 and CCL2 Regulates Invasiveness in Breast Cancer

Evidence indicates that PARP1 can be activated by MAP-kinases [20]. Studies from cerebral neurons and cardiomyocytes indicate synergistic interaction between PARP1 and MAP-kinase pathway as well [21]. In our previous study, we have shown that CCL2 controls breast cancer invasiveness via the ERK1/2 signaling [4]. Hence, we wanted to test whether CCL2 can affect PARP1 activity as well. To this end, we treated MB-231 cells with recombinant human CCL2 (rhCCL2). We observed significant upregulation in total PAR levels upon rhCCL2 treatment by western blot (Figure 5A left). Expectedly, phosphorylation of ERK1/2 was increased upon the addition of rhCCL2. We utilized ERK1/2 phosphorylation as an indication of effective rhCCL2 treatment since it is known to activate the Map kinase pathway. Immunofluorescence with the PAR antibody shows increased PAR levels (green) in rhCCL2-treated MB-231 cells (Figure 5A right). We also observed downregulation in ERK1/(P44/42) phosphorylation with PJ34 treatment indicating inactivation of the MAP Kinase pathway (Figure 5B). Since CCL2 mediated cell signaling pathway activation and PARP1 inhibition are negatively correlated in breast cancer cells, we compared whole transcriptome profiles of MB-231 with rhCCL2 or PJ34 treatment. To analyze the transcriptomic profile between CCL2 and PJ34 treatment, we chose the MAP-kinase pathway as CCL2 is a known activator of ERK1/2. We observed that rhCCL2 and PJ34 treatment show opposite gene expression profiles, particularly in KEGG MAP-kinase pathway associated genes (Figure 5C). IPA analysis with a differentially expressed gene from CCL2 or PJ34 treatment also yielded similar results (Figure 5C right). According to a comparative study between the two conditions on IPA, we observed that CCL2 treatment and PARP inhibition shows the opposite effect on cellular signaling pathway activation or repression (Figure 5C right panel). For example, oxidative phosphorylation and NRF2 mediated pathways are upregulated in response to PJ34 treatment.

In contrast, those pathways are downregulated in CCL2 treated cells. PPAR signaling and PPARα/RXRα activation are upregulated or significantly affected by rhCCL2 treatment. However, those pathways are down-regulated in PJ34 treatment along with ERK5 signaling.

In terms of invasive properties, we found that the addition of rhCCL2 resulted in a moderate rescue of migration in PJ34 treated MB-231 cells (Figure 5D). Cells with PJ34 treatment showed delayed wound closure compared to untreated (Figure 5D top two panels). On the other hand, rhCCL2 treatment partially rescued delayed wound closure (Figure 5D bottom panel).

### 2.7. CCL2 and NFκB Correlate in Breast Cancer Patients Affecting Relapsed Free Survival

To investigate the potential clinical aspects of our findings, we queried publicly available gene expression data via cBioPortal (www.cbioportal.org). We found that *CCL2* mRNA levels are significantly upregulated in Claudin-low type cancer in the METABRIC dataset of 2509 primary breast tumors with 548 matched normal samples (Figure 6A). Claudin-low and Basal subtypes were determined by PAM50 classification and are predominantly TNBCs [22]. However, *P65* did not show any significant transcriptional upregulation except moderate upregulation in the Basal type (Figure 6A right).

Interestingly, *CCL2* and *P65* mRNA expression are positively correlated in the dataset (Figure 6B Spearman *r* = 0.16, *p* < 0.0001 two-tailed). Linear regression analysis also showed significant dependence on CCL2 mRNA expression and P65 mRNA expression as well (Figure 6B right). After that, we queried TCGA pan-cancer normalized dataset from breast cancer patients from KMPLOT for relapsed free survival. A median split was used for *CCL2* and *P65* expression levels to stratify patients in high-expression or low-expression groups. We observed that “*CCL2-P65* high” patients showed a reduction in relapsed free survival compared to “*CCL2-p65* low patients” (Figure 6C, Log-rank Wilcoxon test, *p* = 0.0399). Thus, *CCL2* and *P65* transcriptional upregulation contribute to a worse prognosis for breast cancer patients.

## 3. Discussion

PolyADP ribosylation by PARP1 is a prominent mechanism of protein activity modulation in cancer cells. Recent evidence indicates a broader role for PARP1 that involves chromatin modification, transcriptional mechanisms, and mRNA stability outside of canonical DNA damage response. Here we show that in the context of breast cancer, PARP1 can control inflammatory cytokine production in cancer cells. Focusing on *CCL2* genetic regulation as a PARP1 target, we show that PARP1 interacts with the NFκB P65 subunit, thereby maintaining transcription of *CCL2*. CCL2, in turn, can positively modulate PARP1 activity in cancer cells (Figure 6D).

Interestingly, PARP1 and NFκB mediated mechanisms have been shown to regulate secretory phenotype in senescent cells, which might be necessary for cancer cells to recruit macrophages after chemotherapy treatment and thus may negatively affect the outcome of the procedure [9]. PARP1 is also known to directly interact with both P65 and P50 as a coactivator, which does not require enzymatic activity [19]. However, whether PARP1 mediated PARylation is required for P65 activity remains to be seen. Along with that, PARP1 was previously shown to regulate *CCL2* transcription in NK2 cells [23]. We now show that PARP1 might be functional in multiple ways to regulate chemokines like *CCL2* in breast cancer as well. We found that PARP1 might affect CCL2 promoter by regulating NFκB recruitment as well as itself localizing to the *CCL2* promoter, perhaps modulating chromatin architecture [24].

Moreover, PARP1 is confined throughout the *CCL2* locus, as seen in the Nalabothula et al., Chromatin IP dataset (GSE61916, Appendix A) [25]. Expectedly, *CCL2* transcript coverage in the *CCL2* exons showed a drastic reduction with PJ34 treatment or PARP1 knockdown (Appendix A). PARP1 might also be involved in transcriptional splicing mechanisms, as speculated by Nalabothula et al. (25). In terms of cellular effects, we found that PARP1 inhibition can negatively affect cell proliferation, low-attachment colony formation, as well as cellular migration, which might be due to the downregulation of *CCL2* [26]. Histone PARylation at the promoter, as well as interaction with P300 histone acetyltransferase, might be a potential mechanism to be examined in the future. Along with that, post-transcriptional cytokine mRNA stabilization by PARP1, as reported previously, can be another level of control in the context of breast cancer [27].

We have observed that *CCL2* and, to some extent, *P65* expression are higher in breast cancer patients from the METABRIC dataset. However, we could not determine PARP1 expression as it was not available for the same dataset. In the TCGA breast cancer data, (U.C. Santa Cruz, xenabrowser.net, Nature 2012), two of the factors in this study (*PARP1* and *CCL2*) showed significant overexpression in the Basal type of tumors concerning the Luminal-A or B subtype (Appendix A) [28]. Since *CCL2* or *P65* can be upregulated in infiltrating immune cells, we cannot rule out a contribution from the tumor microenvironment instead of only tumor cells. However, we find that *CCL2*, *PARP1* are highly expressed at least in the Basal-type cancers, the majority of which falls into the TNBC category (Appendix A). It is interesting to note that various PARP1 inhibitors are in clinical trials; however, they have geared towards BRCA1/2 or DNA damage repair deficiency in breast and ovarian cancers [29]. In our study, we provide evidence that the mechanisms of PARP1 action outside of DNA repair could be considered for treatment options for breast cancer patients.

Moreover, small molecule inhibitors against CCL2 cognate receptor CCR2 are in clinical trials mainly in pancreatic cancers [30]. *CCL2* function extends beyond that and shows a significant role in the tumor microenvironment in affecting macrophage and modifying them towards tumor-promoting phenotype. PARP1 inhibition in *CCL2* expressing cancer cells can alter not only the cancer cells themselves but also the surrounding cells in tumor stroma. Thus, it might be an effective combinatorial treatment with PARP1 and CCR2 inhibitors to achieve better outcomes where *CCL2* levels are determined to be high, particularly in breast cancer, as well as other solid tumors.

## 4. Materials and Methods

### 4.1. Cell Cultures, Antibodies, and Reagents

All cell lines were purchased from the ATCC (American Type Culture Collection). MDA-MB-231, BT549, HCC1937, MDA-MB-436, MDA-MB-453, HCC70, HS578T, MDA-MB-468, BT474, T47D, and MCF7 cells were grown in DMEM/F12 medium (Thermo Fisher Scientific, Waltham, MA, USA Cat. No. 10565018) containing 10% FBS, 1mM glutamine and antibiotics (Penicillin/Streptomycin) in 5% CO_2_ incubator at 37 °C. Antibodies used are as follows: PARP1 (Cell Signaling Technology (CST), Gaithersburg, MA, USA, 9542). NFκB Sampler kit containing phospho-NFκB p65 (Ser536) and total NFκB p65 (CST. Cat. No. 9936T). Anti-PAR Monoclonal Antibody (Trevigen, Gaithersburg, MD, USA, Cat no. 4335-MC-100). Goat anti-mouse Alexa fluor Plus 488 (Thermo Fisher Scientific, Cat. No A32723). PJ34 hydrochloride was purchased from Millipore Sigma (Sigma, Burlington, MA, USA, P4365) Leptomycin B was purchased from Millipore Sigma (Cat. L2913). Thiazolyl Blue Tetrazolium Bromide (MTT) was also from Millipore Sigma (Cat. No. M5655). PDD00017273 was from Tocris, Minneapolis, MN, USA (Cat. No. 5952). BAY 11-7082 was from Selleckchem (Houston, TX, USA, Cat. No. S2913). Recombinant Human TNF-alpha Protein (Cat. No. 210-TA-100) and Recombinant Human CCL2/MCP-1 Protein (Cat. No. 279-MC-050) were from R&D Systems, Minneapolis, MN, USA Phospho ERK1/2 and Total ERK1/2 were from CST. (Phospho-p44/42 MAPK (Erk1/2) (Thr202/Tyr204), Cat No 4370 and p44/42 MAPK (Erk1/2), Cat. No. 4695) respectively. GAPDH antibody was from Santa Cruz Biotechnology (Dallas, TX, USA, GAPDH Antibody 6C5, Cat. No. sc-32233)

### 4.2. Whole Transcriptome Analysis

RNAs from treated and control cells were isolated using a miRNeasy RNA isolation kit from Qiagen, Germantown, MD, USA (217004). Truseq mRNA or total Library preparation was performed at Children’s Hospital Los Angeles (Los Angeles, CA, USA). Sequencing libraries were run on Nextseq 500 (Illumina, San Diego, Ca, USA). Data were analyzed with the CLC genomics workbench 12 (version 12, Qiagen, Germantown, MD, USA) with GRCh37 human genome assembly and Ingenuity Pathway Analysis was performed. GSEA was performed with GSEA v3 from Broad Institute in pre-ranked mode against MsigDB databases (MsigDB oncogenic signature). Differential gene expression was calculated in the CLC genomics workbench using the edgrR (Empirical Analysis of Digital Gene Expression Data in R) method.

### 4.3. Enzyme-Linked Immunosorbent Assay (ELISA) for Human CCL2

ELISA was performed twice in duplicate using the Human CCL2 Quantikine ELISA Kit (R&D Biosystem DCP00, Minneapolis, MN, USA) following the manufacturer’s instructions. CCL2 levels (pg/mL) were normalized to 1 × 10^6^ cells.

### 4.4. RNA. Isolation and Quantitative Real-Time PCR

RNA was extracted from sub-confluent cells grown in tissue culture plates with Trizol (Life Technologies, Carlsbad, CA, USA 15596026). RNA was eluted in nuclease-free water. The concentration and purity of the isolated RNA were measured using NanoDrop (Thermo Fisher Scientific, Waltham, MA, USA). Experiments were performed in triplicates and repeated twice. First-strand cDNA synthesis was performed using iScript Reverse Transcription Supermix (Biorad, Hercules, CA, USA 1708840) following the manufacturer’s instructions. The expression levels of CCL2 were tested along with a GAPDH control by qPCR on a CFX96 Touch thermal cycler. The results were analyzed by the 2^−ΔΔCt^ method to show relative expression over the control gene. Primer sequences are as shown below with GAPDH primer form RTPrimer DB.

CCL2-F: CCGAGAGGCTGAGACTAACC

CCL2-R: CTTTCATGCTGGAGGCGAGA

GAPDH-F: TGCACCACCAACTGCTTAGC

GAPDH-R: GGCATGGACTGTGGTCATGAG

### 4.5. Cell Proliferation and Colony Formation Assay

Cells were plated in each well of 96-well plates and assessed for growth inhibition after 48–72 h using the MTT (3-(4,5-Dimethylthiazol-2-yl)-2,5-Diphenyltetrazolium Bromide) assay. For colony formation assay, about 1000 cells/well were seeded into low attachment 48 well plates for 7days, in triplicates with or without PJ34 treatment. Bright-field pictures were taken with a Leica inverted microscope at 10× magnification.

### 4.6. Wound Healing and Boyden Chamber Invasion Assay

A wound-healing assay was performed on a 96-well plate. The wound was created using Biotek (Biotek, Winooski, VT, USA) autoscratcher. The wound area was imaged using Gen5 software on Cytation 1 imager for every hour for 10–12 h after treatment. Wound closing was measured using Gen 5 software (Biotek, Winooski, VT, USA) and graph created using Microsoft Excel. 8-μm pore 24-well Boyden chamber inserts were coated with Matrigel at 1:5 dilution (Corning Biosciences, Corning, NY, USA), and cells were plated on the top chamber. The bottom chamber contained the indicated amount of recombinant human CCL2 (rhCCL2) when used. The number of cells migrated onto the membrane was measured after overnight incubation. Cells were fixed with 4% paraformaldehyde (Diluted with PBS from Fisher Scientific, Cat. No. F79-1 37% Solution) and stained with 0.1% Crystal Violet (Diluted from Millipore/Sigma, Cat. No. V5265 1% aqueous solution). Invasion assays were carried out in duplicates at least twice.

### 4.7. Western Blotting

Total protein was extracted using RIPA buffer (Thermo Fisher Scientific, Waltham, MA, USA, Cat. No. 89901) or Pathscan ELISA cell lysis buffer for ELISA (CST) with 1× Halt protease/phosphatase inhibitor cocktail (Thermo Fisher Scientific). Protein concentration was measured using BCA protein assay (Thermo Fisher Scientific cat 23225). 20–30 µg protein was separated on an 8–10% gel and western blot performed. For multiple protein detection, membranes were stripped in 5 mL of stripping buffer (Restore™ Western Blot Stripping Buffer, Thermo Fisher Scientific, Cat. No. 21059) at 37 °C for 15 min. After three washing with TBS (Tris-buffered Saline), membranes were incubated at room temp with 5% B.S.A. in TBST before proceeding with the next set of antibodies. All phosphoprotein antibodies were done first before stripping and probing with total protein antibodies and GAPDH loading control.

For PARG inhibition, we used Optimem (Thermo Fisher Scientific Cat no. 11058021) with 1% BSA to induce serum starvation to reduce basal PAR levels in the cells. This was followed by treatment with PARGi to inhibit de-PARylation (Figure 3C). For rhCCL2 treatment (Figure 5A), we also used Optimem with 1% BSA to induce serum starvation before the treatment.

### 4.8. Transfection of shRNA/siRNA and Generation of Stable Lines with shRNA Knockdown

BT549 cells were transfected with lentiviral particles (Origene, Rockville, MD, USA TL315488V) at ~1 multiplicity of infection to generate PARP1 knockdown cells, which were utilized for sequencing. siRNAs targeting human PARP1 (Thermo Fisher Scientific s1099 and s1097) and control pool were used to knockdown PARP1 in MB-231 and BT549 cells using Lipofectamine RNAi MAX (Thermo Fisher Scientific Cat no. 13778030) following manufacturers recommendation. PARP1 shRNA transfected BT549 cells were selected under 0.75 µg/mL Puromycin (Puromycin Dihydrochloride, Thermo Fisher Scientific, Cat. No. A1113803) for three weeks to generate stable clones after an overnight lentiviral transfection. Puromycin selection was started 48 h post-transfection.

### 4.9. Co-Immunoprecipitation and Chromatin Immunoprecipitation

Active motif PARP1 N-terminal antibody (Cat no. 39559) was used for Chromatin I.P. Cell Signaling NFκB p65 (D14E12) for p65 pull down from MDA-MB-231 cell lysates. Chromatin I.P. was performed using Millipore/Sigma EZ-Magna ChIP kit (Cat. 17-10086) following the manufacturer’s recommendation. Approximately 1 × 10^6^ cells were used for each I.P. Sonication of cells was performed in a water bath sonicator for 20 s sonication at 20% output with 30 s off-cycle for 3.5 min in ChIP Lysis buffer in the kit.

We harvested formaldehyde-fixed chromatin from the cells and subjected to immunoprecipitation with P65 antibody. Precipitated DNA was analyzed by qPCR with primers flanking the NFκB consensus motif in the CCL2 promoter. NFκB Chromatin IP data reflects % input.

Co-immunoprecipitation of P65 was performed using the P65 antibody using NP40 cell lysis buffer (Thermo Fisher Scientific FNN0021). The following primers for CCL2 were used flanking an NFκB binding motif

CCL2-Forward: GTTCTGCTAGGCTTCTATGA

CCL2-Reverse: GAAAACTGCAGAAAAGGAAG

### 4.10. Gel Shift Assay with NFκB Probes

We utilized Signosis, CA, USA EMSA kit for NFκB (NFκB EMSA GS-0030.) according to manufacturer instructions. Cytoplasmic and nuclear lysates were prepared by Signosis nuclear extraction kit (SK-0001). We utilized the nuclear lysate from PJ34 treated or untreated control cells and incubated with P65 probes. Briefly, 5 × 10^6^ cells were resuspended in 1 mL of Buffer 1 with Dithiothreitol (DTT). After 10 min incubation and centrifugation at 12,000 rpm, 5 min at 4 °C, the supernatant was saved as the cytoplasmic fraction. Pellet was resuspended in 200 µL of Buffer II and shaken for 2 h. The supernatant was collected as a nuclear fraction after centrifugation at 12,000 rpm 5 min at 4 °C. After 1 h incubation, the reaction mixture was separated on a polyacrylamide gel and transferred on to a nitrocellulose membrane. Probe signals were captured using chemiluminescence on a Licor imager.

### 4.11. Immunofluorescence with MB-231 Cells

A total of 5 × 10^4^ MB-231 cells were seeded onto Poly-L-lysine solution (Millipore/Sigma, Cat. No. P4707) coated coverslip. The following day cells were serum-starved overnight. After serum, starvation cells were treated with rhCCL2 for 1 h. Cells were then fixed with ice-cold methanol at −20 °C for 12 min. Anti-PAR antibody was added, followed by anti-mouse Alexa-fluor-488. After washing with Phosphate Buffered Saline (PBS), cells were mounted with Vectashield DAPI and imaged with a Leica SPE confocal microscope (Buffalo Grove, IL, USA).

### 4.12. Annexin V Apoptosis Assay

FITC Annexin V Apoptosis Detection Kit with 7-AAD (Biolegend, San Diego, CA, USA, Cat. No. 640922) was used. MB-231 cells were treated with 10 and 20 µM PJ34 overnight. 5 × 10^5^ Cells were counted, washed with PBS, and resuspended in FITC Annexin V and 7AAD in a 100 µL volume. Cells were resuspended in 500 µL and data acquisition was carried out in an Attune NxT Flow cytometer (Thermo Fisher Scientific, Waltham, MA, USA).

### 4.13. Statistics and Analysis of Publicly Available Data

All experiments performed were at least two independent sets of experiments. Differences between 2 groups were analyzed using Student’s *t*-test or *χ^2^* test in Microsoft Excel. One way ANOVA with Tukey’s test was performed with Graph pad prism v8.4.2 (GraphPad Software, San Diego, Ca, USA). Data are presented as mean values ± standard deviation with statistical significance at *p* ≤ 0.05. TCGA and METABRIC breast cancer data were accessed using MSKCC cBioPortal (www.cbioportal.org) [31] and from Xenabrowser (https://xenabrowser.net) TCGA Breast Cancer (BRCA) dataset [32]. The relapsed-free survival (RFS) for the TCGA pan-cancer dataset was obtained from KMPLOT [33] (www.kmplot.com) and analyzed using Graph pad prism v8.4.2 with one way ANOVA with multiple comparisons and Tukey’s test.

## 5. Conclusions

Breast cancers, particularly TNBCs, show upregulation of CCL2 and overall PARylation with positive feedback controlling CCL2 expression and PARP1 activity. Thus, our work could have the potential therapeutic benefit of targeting TNBCs, which have worse clinical outcomes due to the lack of targeted therapies.

## Figures and Tables

**Figure 1 cancers-12-01317-f001:**
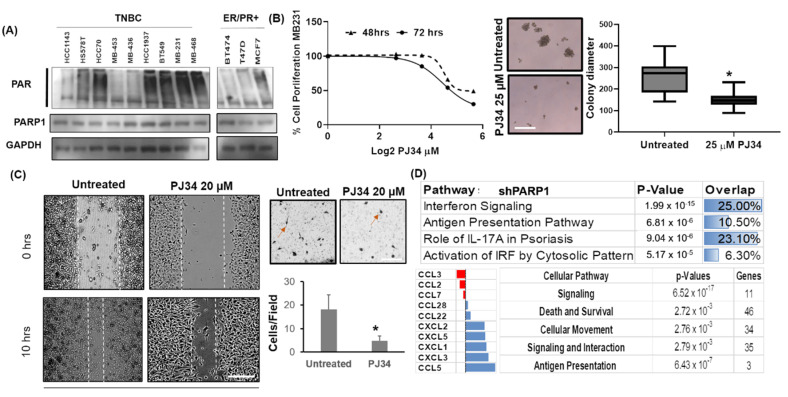
PARP1 inhibition resulted in reduced cell proliferation and migration in breast cancer cells. (**A**) Western blots for total levels of PARP1 and PolyADP Ribose (PAR) in a panel of breast cancer cells. Triple-negative cell lines are on the left. GAPDH is used as a loading control. Whole western blots for 1A are in Appendix A (**B**) Left: Cell proliferation upon PJ34 PARP inhibitor treatment in MB-231 cells at 48 h and 72 h. Right: Colony formation assay with PJ34 treated MB-231 cells in low attachment plates for 7 days. Scale bar 10 µm (**C**) Left: Wound healing assay in the presence of PJ34 in MB-231 cells. Right: Boyden chamber Invasion assay in the presence of PJ34 inhibitor using MB-231 cells. Cells were incubated with PJ34 for the duration of the experiment. Experiments were done in duplicate two times. Dark objects on the pictures are cells. One arrow is shown in each panel, pointing to one of the cells. * *p* < 0.05 × 2 test in (**C**). Scale bar 50 µm (left) and 10 µm (right). Enlarged view in Appendix A. (**D**) Ingenuity pathway analysis (IPA) for Canonical pathways with *PARP1* knockdown in BT549 cells. IPA was used on differentially expressed genes after whole transcriptome sequencing. Overlap showing the % of differentially expressed genes that belong to the pathways shown. Chemokines showed in (**D**) bottom left show similar patterns of expression with PARP1 knockdown. The color bars below indicate downregulation (red) and upregulation (blue) with *PARP1* knockdown. Log2 fold change is represented by the color bars. “Genes” column bottom right showing several differentially expressed genes that belong to the molecular pathway category.

**Figure 2 cancers-12-01317-f002:**
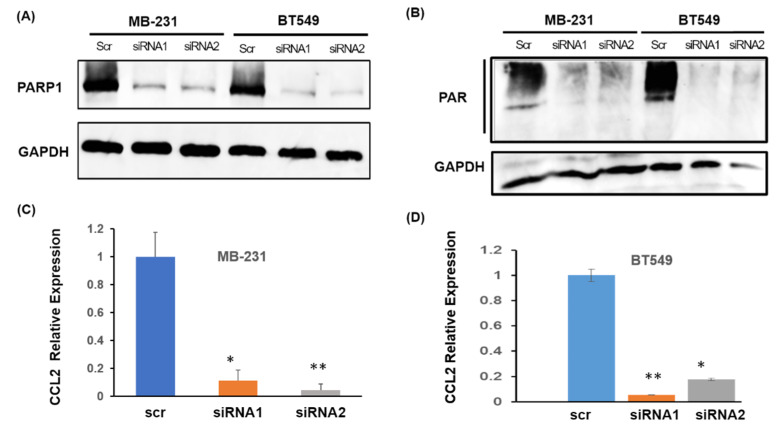
siRNA mediated *PARP1* knockdown downregulates *CCL2* transcription. (**A**) Western blot for Total PARP1. Knockdown efficiency with PARP1 RNAi is confirmed with two different PARP1 siRNAs in MB-231and BT549 cells. (**B**) Western blot showing the total PAR level in the two breast cancer cells with siRNA mediated PARP1 knockdown. GAPDH is shown as a loading control. (**C**) Relative expression of *CCL2* mRNA upon siRNA mediated PARP1 knockdown in MB-231 and (**D**) BT549 cells (Right). For (**C**) and (**D**), RNA was harvested after 72 h post-transfection. siRNA knockdown was performed independently twice. Polymerase chain reaction (PCR) was performed twice in triplicates, data presented as ± S.E.M., * *p* < 0.0004 ** *p* < 0.0001 with one way ANOVA and Tukey’s comparison with scrambled siRNA. Whole western blots for 2A and 2B are in Appendix A.

**Figure 3 cancers-12-01317-f003:**
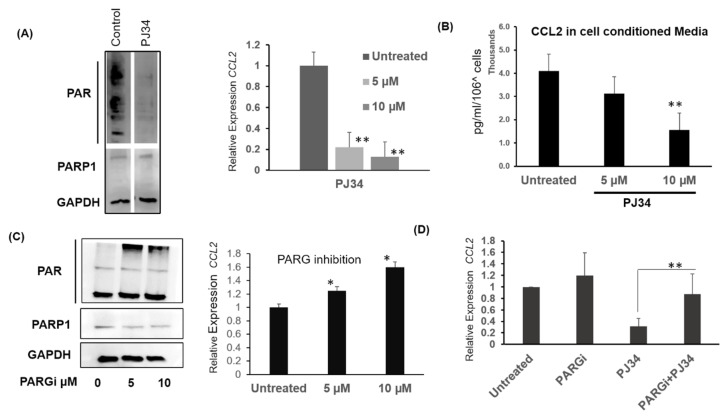
Modulating intracellular PAR levels affects *CCL2* transcription. (**A**) PAR and PARP1 levels after overnight 10 µM PJ34 treatment in MB-231cells. Left: Western bolt showing PARP1 and PAR levels. Right: Relative *CCL2* transcript levels after overnight PJ34 treatment in MB-231with PARP1 inhibitor PJ34. (**B**) Secreted CCL2 (pg/mL) in cell-conditioned media after PJ34 treatment. Serum-free cell-conditioned media was collected and subjected to CCL2 ELISA. CCL2 protein levels were normalized to 10^6^ cells. (**C**) Left: PAR and PARP1 levels after overnight PARG inhibitor (PDD 00017273) treatment in MB-231 cells. Right: Relative expression of *CCL2* mRNA after PARG inhibitor treatment. (**D**) Relative expression of *CCL2* mRNA in the presence of PARG inhibitor, PARP1 inhibitor and the combination of the two at 5 µM and 10µM doses respectively, Data ± S.E.M. ** *p* < 0.0003, for (A) and (C) * *p* < 0.001 one way ANOVA and Tukey’s comparison with untreated, representative of two independent experiments. (D) *χ^2^* test *p*< 0.001. Whole western blots for 3A and 3C are in Appendix A.

**Figure 4 cancers-12-01317-f004:**
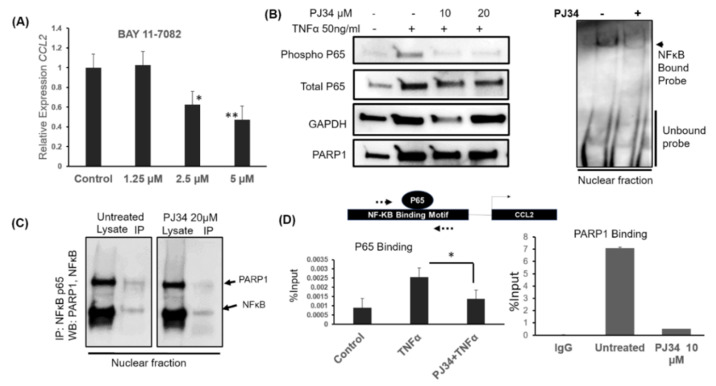
PARP1 and NFκB interaction is important for transcriptional control of *CCL2*. (**A**) Quantitative real-time PCR for relative expression of *CCL2* mRNA after NFκB p65 inhibitor Bay 11-7081 treatment in MB-231 cells. Cells were rhCCL2-treated at the indicated doses for 4 h. * *p* = 0.001, ** *p* = 0.0002, one way ANOVA and Tukey’s comparison with untreated (**B**) Left: Western blot showed activation of NFκB pathway probed with an antibody against phosphorylated P65 S536. MB-231 cells were pretreated with PJ34 overnight, and TNFα was added for 45 min. Right: Electrophoretic mobility shift assay (EMSA) with p65 probe. The nuclear lysate was prepared from PJ34 treated or untreated cells, incubated with p65 specific biotinylated DNA probe. Bound and unbound probes were visualized using HRP-conjugated streptavidin chemiluminescence. (**C**) PARP1 and NFκB interact in MB-231cells. Immunoprecipitation was done with the P65 antibody using nuclear lysate. Lysate and IP fractions were western blotted and probed with PARP1 and P65 specific antibodies. (**D**) Chromatin IP with PARP1 and P65 specific antibody. Fixed chromatin was immunoprecipitated with the antibody, as mentioned earlier, after overnight treatment with 10 µM PJ34. We performed PCR on a region flanking an NFκB binding motif at the *CCL2* promoter (Dotted arrows showing the approximate location of the primers). Percent of input shows enrichment. Average of two independent experiments shown. *t*-test * *p* < 0.01. Whole western blots for 4B and 4C are in Appendix A.

**Figure 5 cancers-12-01317-f005:**
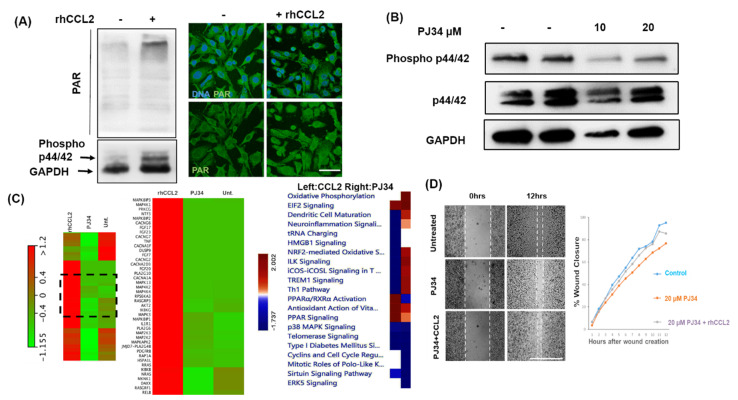
PARP1 and CCL2 crosstalk regulate invasiveness in breast cancer. (**A**) Left: Recombinant human CCL2 (rhCCL2) can increase PAR levels in MB-231 cells. MB-231 cells were serum-starved overnight and treated with 50 ng/mL rhCCL2 for 1 h. Western blot showing total PAR level upon treatment. Right: Immunofluorescence images showing PAR (green) in cells after rhCCL2 treatment. An enlarged image in Appendix A. Merged image with DNA (DAPI) shown on top. Scale bar 20 µm. (**B**) PARP1 inhibitor treatment reduces phosphorylated P44/42 (ERK1/2) levels in MB-231cells. Western blot showing Phospho-p44/42 ERK1/2 (Thr202/Tyr204) and total ERK1/2 after overnight PJ34 treatment followed by 45 min of TNFα. (**C**) Left: Heat map showing transcriptome analysis from MB-231 cells for rhCCL2 (1 h treatment as in **A**) and PJ34 (10 µM Overnight treatment) treated groups. The KEGG MAP kinase pathway genes are identified. Red color indicates upregulation with green, showing the downregulation of mRNA transcription. Right: Ingenuity pathway analysis from rhCCL2 (Left column) and PJ34 treated (right column). Enlarged images of (**C**) are in Appendix A. (**D**) Wound healing assay with simultaneous rhCCL2 and PJ34 treatment on MB-231 cells. Near confluent cells were pretreated with PJ34 before rhCCL2 was added at the dose of 50 ng/mL after wound creation. Treatment was continued for 12 h. The graph on the right shows % wound closure with respect to time. Scale bar 100 µm. Whole western blots for 5A and 5B are in Appendix A.

**Figure 6 cancers-12-01317-f006:**
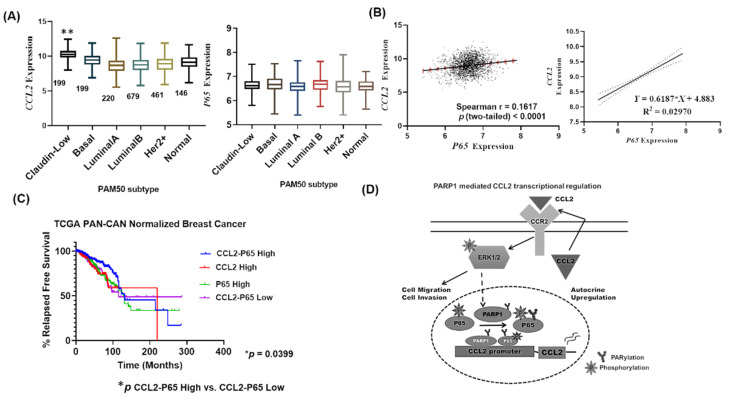
*CCL2* and *P65* mRNA expression are positively correlated in breast cancer and affect relapsed free survival (**A**) METABRIC Dataset showing *CCL2* (Left) and *P65* (Right) mRNA expression as determined by microarray analysis. Breast cancer patients are classified based on the PAM50-Claudin Low signature on the horizontal axis. The numbers in the figure represent the total number of patients in each group. *CCL2* expression is significantly higher in the Claudin-low type (One way ANOVA followed by Tukey’s multiple comparisons. ** *p* < 0.001 vs. Normal). (**B**) Left: Correlation of *CCL2* and *P65* mRNA expression in patients. Spearman correlation with a two-tailed test for *p*. Right: Linear regression showing an association between *P65* and *CCL2* mRNA expression. Dotted lines were showing confidence bands representing the confidence interval of the best fit line (solid). (**C**) Kaplan-Meier survival curve showing relapsed free survival (RFS) in patients categorized based on high and low (median split) *CCL2* and *P65* expression. TCGA breast cancer patients from KMPLOT. Log-Rank Wilcoxon *p* for *CCL2-P65* High vs. Low groups. (**D**) Model of PARP1 and CCL2 pathways interaction upregulating invasiveness in breast cancer. CCR2 is shown as the cognate receptor for CCL2. CCL2 binds to CCR2 and activates MAP Kinases, which results in the downstream activation of PARP1 and P65, followed by their recruitment to the *CCL2* promoter.

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
