# Peer review of "Transcriptional Regulation of CCL2 by PARP1 Is a Driver for Invasiveness in Breast Cancer"

_cancers, 2020, doi:10.3390/cancers12051317_

Round 1
Reviewer 1 Report
Dutta et al. have researched the important aspects of cytokine signaling in triple-negative breast cancer (TNBC). The authors have deployed a wide range of molecular biological and genetic tools to elegantly demonstrate that PARP1 modulates chemokine ligand 2 (CCL2) activity by directly polyADP-ribosylating p65. Importantly, this pathway is proposed to sustain itself in a positive feedback loop thereby constituting a vicious cycle that may act in an autocrine or paracrine manner on neighboring tumor or stromal cells. In addition, the authors went on to corroborate their findings by querying web-based platforms to show that indeed there is a correlation between PARP1-dependent p65 signaling and poor prognosis in breast cancer patients. Although these novel conclusions may be key to the design and guidance of future chemotherapeutic strategies for TNBC, the manuscript heavily suffers from critical mistakes made during the preparation and assembly of scientific evidence (see the major points below).
Major points:
1) There seems to be a problem in Figure 1A (line 87) in that PARP1 western blot images don't exactly correspond to what is shown in Figure S7 for triple-negative and perhaps even for the other three cell lines (BT474, T47D, and MCF7) as the background does not turn white in the lower left cut-out corner side of the PARP1 image presented in the supplementary file. Please correct and double-check that the same images are shown between the cropped and uncropped versions for other western blots.
2) GAPDH western blot images are almost identical between Figures 5B and Supplementary figure 5C. Please revise your western blot results thoroughly with increased attention to the way these data are documented, organized, assembled, and archived.
3) Nuclear fractionation assay presented in Supplementary figure 5A lacks control markers for individual cell compartments that would demonstrate purity of the isolated fractions and at the same time would serve the purpose of loading controls to show constant expression between the PJ34-treated and untreated groups. Please supplement the figure with these important controls.
Minor points:
1) Please define abbreviations for "IRF" (line 105), "EDGR" (line 115), "EMSA" (line 214).
2) PARP1 is misspelled as PAPR1 (lines 16, 20, 67, 259, 279, 397, Supplementary figure 3B, Supplementary figure 4, Supplementary methods - Cell cultures, antibodies and reagents, Supplementary methods - Transfection of shRNA/siRNA and generation of stable lines with shRNA knockdown 2x).
3) "triple-negative" (line 19), "promoter" (lines 25, 203, 229, 233, 236, 257, 340, 341, and 348), "treatment" (lines 27 and 416 and Supplementary figure 3 2x), "kinase" (line 63), "invasion assay" (lines 81 and 93), "white" (line 83), "canonical" (line 94), "signaling" (line 103), "polycomb" (line 128), "not" (line 135), "poly" (line 174), "glycohydrolase" (line 174), "experiments" (line 197), "polyacrylamide" (line 216), "chromatin" (lines 231, 233, 237, 240, 241, 255, 342, Supplementary figure 5 2x, and Supplementary methods - Co-immunoprecipitation and chromatin immunoprecipitation), "green" (lines 266 and 282), "relapsed free survival" (line 299), "dataset" (line 309), "low" (line 317), "log-rank" (line 326), "transcriptome analysis" (line 376 and Supplementary methods - Whole transcriptome analysis), "colony" (line 385 and Supplementary methods - Cell proliferation and colony formation assay 2x), "using" (line 387), "chamber" (line 389 and Supplementary methods - Wound healing assay and Boyden chamber invasion assay), "chromatin immunoprecipitation" (line 399 and Supplementary methods - Co-immunoprecipitation and chromatin immunoprecipitation), "inflammatory and oncogenic" (line 414 and Supplementary figure 1), "gene" (line 415 and Supplementary figure 2), "chromatin and downregulated" (line 418 and Supplementary figure 5), "heatmap" (Supplementary figure 2), "in" (Supplementary figure 3), "chromosome" (Supplementary figure 4), "glutamine" (Supplementary methods - Cell cultures, antibodies and reagents), "pictures" (Supplementary methods - Cell proliferation and colony formation assay) should all be spelled with the first letter(s) lower-case.
4) Please replace "Co-Immunoprecipitation" with "Co-immunoprecipitation" (line 399 and Supplementary methods - Co-immunoprecipitation and chromatin immunoprecipitation 2x).
5) The statement "to regulate transcription CCL2" (line 24) is not grammatically correct. Please replace with "to regulate CCL2 transcription" or "to regulate transcription of CCL2"
6) Please remove period from "upon CCL2. Treatment" (line 27).
7) "mechanism" could be in plural form "mechanisms" (line 46).
8) Please swap "MB-231" (line 73) and "MDA-MB-231 (MB-231)" (line 75) as the abbreviation for MDA-MB-231 cells should be mentioned at the place of its first occurrence (line 73).
9) The label "(Figure 1B Left panel)" should read "(Figure 1B Right panel)" (line 79).
10) There is a discrepancy between PJ34 concentration used in the Boyden chamber invasion assay between the text (20 uM) (line 84) and Figure 1C (25 uM) (line 87).
11) There is another problem in Figure 1A (line 87) as the number of samples probed (9) is different from the number shown as GAPDH loading controls (10). Please recrop the GAPDH image so that it better corresponds to what is shown in Figure S7.
12) The last issue about Figure 1A (line 87) is that there are two small unwanted marks around the square framing the PAR image part (on the outer edge of the upper right corner side just below the MB-231 and MB-468 labels). Although this is just a cosmetic detail, the western blot panel would look graphically more professional without these two redundant spots.
13) Please equip one of the two micrographs in Figure 1B (line 87) with a scale bar.
14) Please equip one of the four micrographs in Figure 1C (left) (line 87) with a scale bar.
15) Due to poor resolution and/or small image size it is very hard to distinguish cells present on the Boyden chamber invasion assay micrographs in Figure 1C right (line 87). Would it please be possible to increase the resolution or provide enlarged versions of these two (treated and untreated) images so that readers can see what is and what is not a cell. Alternatively, enlarged images can be included in the supplementary file.
16) Please equip one of the two micrographs in Figure 1C right (line 87) with a scale bar.
17) Please indicate for how long were MB-231 cells treated with PJ34 inhibitor in the Boyden chamber invasion assay in Figure 1C right (line 87) and in the dose-response transcript level measurement in Figure 3A right (line 190).
18) It is not clear in Figure 1D (line 87), which part represents PARP1-depleted BT549 cells and which part is for PJ-34-treated MB-231 cells. Please indicate this clearly and/or split the panel into two (Figure 1D and new Figure 1E). Organize figure legend accordingly.
19) Please provide more detailed description in the legend to Figure 1D (line 87) about the meaning of the presented data:
a) What does "Overlap" mean?
b) What does red and blue color coding indicate?
c) What does "Genes" column tell?
20) Please make cytokine P values fully visible in Figure 1D (line 87). Also, the colon ":" character in "Pathway :" in the upper left corner seems to be cut through.
21) Although the authors claim that "PARP1 knockdown showed a significant effect on cytokine signaling" (line 101), "In the Ingenuity Pathway (IPA) analysis, interferon Signaling and IL-17A were significantly affected" (line 102), and "PARP1 inhibition modulated inflammation-related pathways" (line 105), from the entire paragraph is not clear what the trend was. Please make it clear in the text whether these pathways upregulated or downregulated following PARP1 inactivation?
22) The sentence "Other routes, such as antigen presentation and activation of IRF, also suggested that PARP1 inhibition modulated inflammation-related pathways." (line 105) is rather confusing as antigen presentation pathway and activation of IRF by cytosolic pattern are in the shPARP1 column. Please reserve the word inhibition for chemical intervention only or indicate for which particular case was PJ34 treatment used in Figure 1D (line 87).
23) Please replace "Ingenuity Pathway (IPA) analysis" with "Ingenuity Pathway Analysis (IPA)" (line 102).
24) "Ingenuity Pathway Analysis (IPA)" (lines 106) and "Ingenuity Pathways (IPA) analysis" (line 272) could be shortened to "IPA".
25) The authors state that "Ingenuity Pathway Analysis (IPA) showed that the significant pathways upregulated in MB-231 are the EIF2 signaling with 42% overlap, oxidative phosphorylation showing 107 49% overlap with the genes of those pathways Supplementary Figure 1). Protein translation-related EIF4 and p70S6K pathways were also positively regulated. Along with that, Sirtuin signaling showed significant changes." (line 106), however there is no mention of EIF2, oxidative phosphorylation (OXPHOS), EIF4, or p70S6K in Supplementary Figure 1. Please modify this figure to clearly contain all this factors, provide tabulated source data, or indicate as "(data not shown)".
26) The formulation of the sentence "A total of 1059 genes were differentially expressed after utilizing EDGR for differential gene expression" (line 114) could be improved as:
a) The number of genes with altered expression following shRNA-mediated PARP1 depletion 1059 is different from 1095, which is indicated in Supplementary Figure 1.
b) This sentence does not mention the fact that this number refers to shRNA-mediated PARP1 depletion, which is confusing because this is being compared to PJ34 treatment in the following sentence: "On the other hand, PJ34 treatment significantly affected the transcriptional status of nearly 5759 genes" (line 115).
27) Please replace "Broad" with "Broad Institute" (page 121).
28) There is a difference in the name of the two genes (CXCL13 and ADMA8) noted in the text as the outcome of the leading edge analysis (line 127) and in Supplementary figure 2 (CXCL3 and ADAM8). Please fix.
29) Please split "survival movement" using a comma as these are two different processes (line 136).
30) Please correct "(Figure & D)" to "(Figure 2C,D) (line 148).
31) The y axes for graphs in Figure 2C & D (line 152) don't have exactly the same height, which makes comparison between MDA-MB-231 and BT-549 cells less accurate.
In addition, the font seems to be different between these two graphs as well. Please replot data using consistent size and text font.
32) The amount of secreted CCL2 1900pg/ml mentioned in the text (line 169) does not exactly correspond to what is shown in Figure 3B (line 187) as from the graph the mean value seems to be less than 1900pg/ml by visual inspection.
33) The statement "We found an 80% to 90% reduction in CCL2 transcript with PJ34 treatment in those two cell lines" (line 172) seems to be incorrect in that MDA-MB-436 cells don't reach such profound reduction following PJ34 treatment when consulting Supplementary figure 3A. Please use increased precision in gauging values from plotted figures.
34) Reference to Supplementary figure 3B is missing. Please incorporate result description of Supplementary figure 3B into the main text.
35) Please remove one of the two sentences: "Interestingly, treating MB-231 cells with PDD 17273 (PARGi) resulted in increased PARylation (Figure 3C) [13]. Expectedly, total PAR levels increased after treatment with PARGi (Figure 3C Left panel)" (line 176) as they both inform on the same effect of increased PAR levels (PARylation) following PARG inhibitor treatment and hence are mutually redundant.
36) Please replace "PDD 17273" with "PDD00017273" (line 177).
37) Please add the word "treatment" after "PARGi and PJ34" (line 183).
38) Please realign western blot images for PAR and PARP1/GAPDH in Figure 3A (line 187) so that there is a visible vertical gap between the untreated and PJ34-treated western blot results for all the target proteins probed.
39) Please remove the small black redundant "minus" mark, which is just above the (A) label in Figure 3A (line 187).
40) Please indicate for how long were MB-231 cells treated with PARG inhibitor in Figures 3C and 3D (line 187) in their respective legends.
41) Remove "Right: Western bolt showing PARP1 and PAR levels." from the legend to Figure 3A (line 189).
42) Remove "(left)" from the legend to Figure 3A (line 191).
43) Remove "is on the right" from the legend to Figure 3B (line 191) as there is only one panel.
44) Change "after PARG inhibitor, PARP1 inhibitor, and in the combination" to "in the presence of PARG inhibitor, PARP1 inhibitor, and the combination" (line 195).
45) Please replace "immune precipitation" with "immunoprecipitation" (line 222).
46) Please replace "(Supplementary Figure 4B)" with "(Supplementary Figure 5B)" (line 226).
47) Reference to Supplementary figures 5C and 5D are missing. Please incorporate result description of Supplementary figures 5C and 5D into the main text.
48) Please replace "treatment P65, specific" with "treatment, P65-specific" (line 226).
49) "Since we observed that NFκB and PARP1 cooperation" is not grammatically correct (line 228). Please correct.
50) Please replace "(Supplementary Figure 4A)" with "(Supplementary Figure 5A)" (line 241).
51) Could the authors briefly comment on in the text as to why there is a decrease observed in the p65 signal in the nuclear fraction of MB-231 cells following PJ34 treatment in Supplementary figure 5A but not in Figure 4C?
52) Please indicate for how long were MB-231 cells treated with PJ34 inhibitor in the p65 and PARP1 antibody chromatin immunoprecipitation assay and what was the dose of PJ34 used with p65 antibody chromatin immunoprecipitation in Figure 4D (line 244).
53) Please state what do the two dashed arrows drawn along the NFκB-binding motif indicate in the legend to Figure 4D (line 244).
54) There seems to be discrepancy between the serine amino acid residue number targeted by the phospho-p65 antibody: The sentence "Interestingly, in the presence of the PARP1 inhibitor p65 serine 436 phosphorylation, associated with NFκB activation, was reduced significantly without affecting the total level of p65" (line 210) refers to Ser436 in contrast to the sentence "Western blot showing activation of NFκB pathway probed with an antibody against phosphorylated p65 S536" (line 247) which points to Ser536. Please remove any ambiguities.
55) Replace "streptavidin HPR conjugated" with "HRP-conjugated streptavidin" (line 252)
56) Replace "showing" with "showed" (line 266).
57) Replace "treated" with "rhCCL2-treated" (line 266).
58) Please incorporate a sentence commenting on the result relevant to the phospho-ERK1/2 western blot in Figure 5A left (line 278) into the main text.
59) Please rescale the four micrographs shown in Figure 5A right (line 278) so that fine morphological details can be discerned inside cells. Alternatively, provide enlarged images as part of the supplementary file.
60) Please equip one of the four micrographs shown in Figure 5A right (line 278) with a scale bar.
61) Reference to Figure 5B is missing. Please incorporate result description of Figure 5B (line 278) into the main text.
62) Please add the original western blot image for GAPDH as part of Figure 5B (line 278) into the supplementary file among other western blot images.
63) Please provide tabulated data underlying the transcriptome analysis presented in Figure 5C (line 278) as an additional supplementary file so that all genes names are visible to the reader.
64) Please provide more detailed description in the legend to Figure 5C left (line 278) as to what does red and green color coding indicate.
65) The first and last gene names present in the zoomed heat map record are cut through in Figure 5C left (line 278). Please correct.
66) Please remove the small intelligible gene description (left) as well as the small intelligible heat map legend (bottom) from Figure 5C left (line 278).
67) Please indicate for how long and using what dose were MB-231 cells treated with rhCCL2 and PJ34 inhibitor in Figures 5C left and right (line 278).
68) Please make IPA pathway names fully visible in Figure 5C right (line 278).
69) "Activation z-score", "-1.203", and "0.533" descriptors are difficult to read because the font color is gray in Figure 5C right (line 278). Would it be possible to change the font color to black?
70) Please add length information for the scale bars displayed in Figure 5D left (line 278) to the respective legend. Also, please remove the small intelligible symbols from above all scale bars in the same figure.
71) Graph "20 uM PJ34" and "20 uM PJ34 + rhCCL2" legends in Figure 5D right (line 278) are plotted using indifferent font colors. Please replot the respective graph legend using orange font color.
72) Please indicate for how long were MB-231 cells treated with rhCCL2 in Figure 5D (line 278).
73) Please distinguish "Left:" and "Right:" sections for the western blot and immunofluorescence result, respectively, in the legend to Figure 5A (line 279).
74) Replace "rhCCL2 and PJ simultaneously on" with "simultaneous rhCCL2 and PJ34 treatment" (line 288).
75) The sentence "PPAR signaling and PARG alpha, ERK5 pathways are upregulated or significantly affected by rhCCL2 treatment" (line 291) is not fully correct as PARG alpha seems to be absent from the IPA list in Figure 5C right (line 278). Please include additional data into Figure 5C, provide tabulated source data, or indicate as "(data not shown)" for PARG alpha.
76) Replace "Cbioprtal" with "cBioPortal" (line 301).
77) Although the authors claim that "We found that CCL2 levels are significantly upregulated in Claudin-low and Basal type cancer in the METABRIC dataset of 2509 primary breast tumors with 548 matched standard samples" (line 301), only Claudin-low data set is indicated as significant in Figure 6A left (line 307). Correct the abovementioned sentence or indicate proper statistical significance in the respective figure using asterisk(s).
78) Similarly, correct the statement for p65: "However, P65 did not show any significant transcriptional upregulation except moderate upregulation in the Basal type" (line 304) or indicate statistical significance in Figure 6A right (line 307) using asterisk(s).
79) In addition, indicate the meaning of numbers "199", "199", "220", "679", "461", "146" to the legend of Figure 6A left (line 307).
80) Could the authors please make it more clear in the text what is the difference between Figure 6B left and right (line 307)? Both figures refer to mRNA expression in the graph but only Figure 6B right but not left is described as such in the text: "Interestingly, CCL2 and P65 expression are positively correlated in the dataset" (line 319). Please indicate CCL2 and p65 expression as "mRNA expression" in this sentence, if mRNA expression is indeed being plotted in Figure 6B left. Please use reasonable amount of stringency when describing experimental data sets.
81) Please indicate what are the two dotted lines in the legend to Figure 6B right (line 307).
82) Please replace "*p CCL2-P65 High with Low" with "*p CCL2-P65 High vs. Low" in Figure 6C (line 307).
83) Please include definition for CCR2 in the legend to Figure 6D (line 307).
84) Swap "CCL2 (Right) and P65 (left)" with "CCL2 (left) and P65 (right)" (line 309).
85) Replace "PAM50-Clausdin Low" with "PAM50-Claudin low" (line 310).
86) The sentence "RELA/P65 expression is higher in basal (Turkeys multiple comparison. ** p < 0.001 with normal) and Claudin-low type" (line 312) is confusing because only Basal but not Claudin-low type are considered significant in the sentence "However, P65 did not show any significant transcriptional upregulation except moderate upregulation in the Basal type" (line 304) and in Figure 6A right (line 307). Please revise both sentences and if p65 Claudin-low data set is indeed significantly different from Normal, indicate this fact in the respective figure using asterisk(s).
87) Replace "basal" with "Basal" (line 312) and "with normal" with "vs. Normal" (line 312).
88) Replace "Turkeys" with "Tukey's" (line 312 and Supplementary figure 6).
89) Please indicate CCL2 and p65 expression as "mRNA expression" (2x) in the sentence "Left: Correlation of CCL2 and P65 in patients. Spearman correlation with a two-tailed test for P. Right: Linear regression between P65 expression and CCL2 expression" (line 313), if mRNA expression is indeed being plotted in Figures 6B left and right (line 307).
90) Replace "Kaplan Myer" with "Kaplan-Meier" (line 314).
91) Replace "kmplot" with "KMPLOT" (line 316).
92) Please replace "r 0.16" with "r = 0.16" (line 320).
93) The classification formulated as "classify a patient as high expressing with the low expression for each of the genes" (line 323) is not clear. Please rephrase.
94) Could the authors briefly comment in the text on the fact that the relapsed free survival was shorter for the CCL2 high group when compared to the CCL2-p65 group?
95) Replace "contribute" with "contributes" (line 326).
96) Could the authors please add a section about whether the interaction between PARP1 and p65 has been previously observed in other studies in the Discussion section?
97) Replace "senescence" with "senescent" (line 337)."
98) Replace "examine" with "be examined" (line 349).
99) Replace "(Supplementary Figure 5)" with "(Supplementary Figure 6)" (line 356).
100) Replace "cells sequenced" with "cells was sequenced" (line 377).
101) Remove the latter "in duplicate" from the sentence "ELISA was performed twice in duplicate using the Human CCL2/CCL2 Quantikine ELISA Kit 379 (R&D Biosystem DCP00) in duplicate" (line 379 and Supplementary methods - ELISA for human CCL2) and please change "CCL2/CCL2" to "CCL2/MCP-1" in the same sentence.
102) Please add "cells" after "BT-549" (line 397).
103) Please remove "and" or fill in the missing word(s) from the sentence "PARP1 siRNA and was used to knockdown PARP1 in MB-231 and BT-549 cells" (line 397).
104) Please provide legend to Supplementary figure 1. Namely, is not clear:
a) What is displayed on the x axis in Supplementary figure 1B?
b) What does red and blue color coding indicate?
c) What do the black vertical lines adjoining the red-blue color map mark?
d) What do the small red marks under the x axis of the plotted graph for IL2_UP.V1_UP in Supplementary figure 1B represent?
e) What is the meaning of and/or abbreviation definition for size, ES, and NES in Supplementary figure 1B?
105) Please provide color legend for the heatmap values in Supplementary figure 2.
106) The statement "The gene sets on the shows enrichment" is not gramatically correct in the legend to Supplementary figure 2.
107) Please replace "genes listed on the column" with "genes listed in the column" in the legend to Supplementary figure 2.
108) Please add error bars to Supplementary figure 3B and indicate if any treatments were made to HCC1937 cell prior to PARP1 immunoprecipitation in the respective legend.
109) Please remove the small unnecessary "minus" mark in Supplementary figure 4, which is on the right side above the purple MCF7 dataset.
110) The label "shPARP1" is cut through in Supplementary figure 4. Please fix.
111) Please change "cell" and "Cell" to "cells" in the legend to Supplementary figure 4 and in Supplementary methods - Co-immunoprecipitation and chromatin immunoprecipitation.
112) Please provide original western blot images for all targets probed in Supplementary figures 5A–D into the supplementary file among other western blot images.
113) Please indicate for how long were MB-231 cells treated with PJ34 inhibitor for each of Supplementary figures 5A–D.
114) It is not clear why IKKα (and not IKKα/β) served as the total signal for comparison in Supplementary figure 5D (when phospho-IKKα/β was used)?
115) Please indicate statistical significance in Supplementary figure 6 using asterisk(s).
116) Please comment on briefly as to the definition of "Basal-Like" tumor types in the legend to Supplementary Figure 6.
117) Please label the panel demonstrating CCL2 expression as "PAM50 Subtype" in Supplementary figure 6 for increased consistency with other panels.
118) Replace "All Cases" with "In all cases," in the legend to Supplementary figure 6.
119) Please provide a brief comment on the meaning of the row of numbers (e.g. 8.5, 9.4, 5.7), in both Supplementary figure 7 and 8, that appear along some of the western blot images.
120) Please relabel "Western blots of Figure 3D" to "Western blots of Figure 3C" in Supplementary figure 7.
121) Please mark the lower set of bands as GAPDH using a red arrow in the PARP1 western blot belonging to Figure 3C in Supplementary figure 7. In addition, the PARP1 label seems to be cut through in the same western blot image. Please fit the label in.
122) Please relabel "Western blot of Figure 4C" to "EMSA result of Figure 4B" and "Western blots of Figure 4D" to "Western blot of Figure 4C" in Supplementary figure 8.
123) Please provide catalog number for the DMEM/F12 medium used for cell cultures in Supplementary methods - Cell cultures, antibodies and reagents.
124) Please provide culture conditions for the following cell lines used: HCC1143, HS578T, HCC70, MB-453, MB-468 in the main text (line 374) and in Supplementary methods - Cell cultures, antibodies and reagents.
125) Please indicate antibody including its catalog number used for PAR, GAPDH, phospho-p65, p65 (total), phospho-p44/42, p44/42 (total), N-cadherin, vimentin, phospho-IKKα/β, and IKKα (total) in Supplementary methods - Cell cultures, antibodies and reagents.
126) Please replace "Sample" with "Sampler" in Supplementary methods - Cell cultures, antibodies and reagents.
127) Please add PDD 00017273, BAY 11-7082, TNFα, rhCCL2 and their manufacturer information into Supplementary methods - Cell cultures, antibodies and reagents.
128) Please remove "Veliparib (ABT-888, Selleckchem S1004)" from Supplementary methods - Cell cultures, antibodies and reagents as this chemical was not used throughout the manuscript.
129) Please change "RNA from treated or control cells were" to "RNA from treated or control cells was" in Supplementary methods - Whole transcriptome analysis.
130) Please change "mirNeasy" to "miRNeasy" in Supplementary methods - Whole transcriptome analysis
131) "instruction" could be in its plural form "instructions" in Supplementary methods - RNA isolation and quantitative real-time PCR and in Supplementary methods - Gel shift Assay with NFκB probes.
132) Please replace "CFX96 touch thermal cycler" with "CFX96 Touch thermal cycler" in Supplementary methods - RNA isolation and quantitative real-time PCR.
133) Please indicate "Cell proliferation and colony formation assay:" in Supplementary methods in bold font.
134) Please change "and assessed for growth inhibition after 72 hrs" to "and assessed for growth inhibition after 48–72 hrs" in Supplementary methods - Cell proliferation and colony formation assay.
135) There seems to be a discrepancy between the amount of cells seeded between the statement "For Colony formation, 5000 cells were seeded in low attached 24 well plates in triplicates with or without PJ34 treatment" (Supplementary methods - Cell proliferation and colony formation assay) and the main text "For colony formation assay, about 1000 cells were seeded into low attachment 48 well plates for 7days" (line 387). Please fix and indicate for how long the cells were seeded for also in the Supplementary methods section statement.
136) There is also discrepancy between the time used for measurement in the wound healing assay since the statements "Wound healing assay was performed on 96-well plate and wound was created using Biotek Autoscratcher and wound area was imaged using Gen5 software on Cytation 1 imager for every hour for 16 hours after treatment mentioned" (Supplementary methods - Wound healing assay and Boyden chamber invasion assay) and "The wound was created and imaged for every hour for 10-16 hours after the treatment mentioned" don't exactly match (line 390).
137) Please add the following instruction "The number of cells migrated onto the membrane was measured after overnight incubation." (line 392) also to the Supplementary methods section - Wound healing assay and Boyden chamber invasion assay.
138) Please indicate how different fractions were prepared for Supplementary figure 5A in Supplementary methods section?
139) Please indicate the method of how successive western blot targets, such as the GAPDH loading control, were processed in Supplementary methods - Western blotting. For example, were western blot membranes stripped before reprobing? If yes, how?
140) Please replace "Cell Signaling technologies" with "Cell Signaling Technology" and "ThermoScientific" with "Thermo Fisher Scientific" in Supplementary methods - Western blotting.
141) Please provide manufacturer for Optimem in Supplementary methods - Western blotting.
142) Please change "Thermofisher" and "ThermoFisher" to "Thermo Fisher Scientific" in Supplementary methods - Western blotting and in Supplementary methods - Co-immunoprecipitation and chromatin immunoprecipitation.
143) Please change also "manufacture’s" to "manufacturer’s" in Supplementary methods - Western blotting.
144) Please describe the procedure for generating stable lines in Supplementary methods - Transfection of shRNA/siRNA and generation of stable lines with shRNA knockdown.
145) Please replace "17010086" with "17-10086" as the correct catalog number for the EZ-Magna ChIP A/G chromatin immunoprecipitation kit in the main text (line 400) and Supplementary methods - Co-immunoprecipitation and chromatin immunoprecipitation.
146) Please change "Cell Signaling" to "Cell Signaling Technology" in Supplementary methods - Co-immunoprecipitation and chromatin immunoprecipitation.
147) Please replace "manufactures" with "manufacturer’s" in Supplementary methods - Co-immunoprecipitation and chromatin immunoprecipitation.
148) Please change "Sonication of Cell were" to "Sonication of cells was" in Supplementary methods - Co-immunoprecipitation and chromatin immunoprecipitation.
149) Please replace "output with 30S" with "output with 30s" in Supplementary methods - Co-immunoprecipitation and chromatin immunoprecipitation.
150) Please change "Following primers for CCL2 was" to "The following primers for CCL2 were" in Supplementary methods - Co-immunoprecipitation and chromatin immunoprecipitation
151) Please replace "standard deviation with statistically" with "standard deviation with statistical" in Supplementary methods - Statistics and Analysis of publicly available data.
152) Please change the web address "www.cbiopotal.org" to "www.cbioportal.org" in Supplementary methods - Statistics and Analysis of publicly available data.
153) Please describe methodology behind the immunofluorescence experiment in Figure 5A right (line 278) in the Supplementary methods section.
Author Response
Reviewer 1 comments:
Comments and Suggestions for Authors:
Dutta et al. have researched the important aspects of cytokine signaling in triple-negative breast cancer (TNBC). The authors have deployed a wide range of molecular biological and genetic tools to elegantly demonstrate that PARP1 modulates chemokine ligand 2 (CCL2) activity by directly polyADP-ribosylating p65. Importantly, this pathway is proposed to sustain itself in a positive feedback loop thereby constituting a vicious cycle that may act in an autocrine or paracrine manner on neighboring tumor or stromal cells. In addition, the authors went on to corroborate their findings by querying web-based platforms to show that indeed there is a correlation between PARP1-dependent p65 signaling and poor prognosis in breast cancer patients. Although these novel conclusions may be key to the design and guidance of future chemotherapeutic strategies for TNBC, the manuscript heavily suffers from critical mistakes made during the preparation and assembly of scientific evidence (see the major points below).
Response: We are grateful to the reviewer of the encouraging comments with the detailed review. We have addressed all comments in the revised manuscripts with changes in the text, figures with updated references.
Major points:
1) There seems to be a problem in Figure 1A (line 87) in that PARP1 western blot images don't exactly correspond to what is shown in Figure S7 for triple-negative and perhaps even for the other three cell lines (BT474, T47D, and MCF7) as the background does not turn white in the lower left cut-out corner side of the PARP1 image presented in the supplementary file. Please correct and double-check that the same images are shown between the cropped and uncropped versions for other western blots.
Response: Once cell line that was not included in the final figure causing the issue. We have corrected the PARP1 blots show the correct bands corresponding to the cell lines labeled on top.
2) GAPDH western blot images are almost identical between Figures 5B and Supplementary figure 5C. Please revise your western blot results thoroughly with increased attention to the way these data are documented, organized, assembled, and archived.
Response: The same blot was stripped and probed with other antibodies. We have included the details about the procedure in the Materials and Methods section.
3) Nuclear fractionation assay presented in Supplementary figure 5A lacks control markers for individual cell compartments that would demonstrate purity of the isolated fractions and at the same time would serve the purpose of loading controls to show constant expression between the PJ34-treated and untreated groups. Please supplement the figure with these important controls.
Response: We have added new nuclear cytoplasmic fraction figure with GAPDH as control. Since the previous figure was done without control antibodies, we remove the figure. In the current figure, we are showing reduction in the nuclear fraction of p65. Chromatin fraction was not carried out for this experiment. We have removed other panels as well due to not being used in the text(C, D) and unavailability of blots (B).
Minor points:
1) Please define abbreviations for "IRF" (line 105), "EDGR" (line 115), "EMSA" (line 214).
Response: We have included IRF as Interferon Regulatory Factors and Electrophoretic mobility shift assay for EMSA. For edgeR, we have included the reference. The full form of edgeR is “Empirical Analysis of Digital Gene Expression Data in R”, which is included in the “Whole transcriptome analysis” section of Materials and Method.
2) PARP1 is misspelled as PAPR1 (lines 16, 20, 67, 259, 279, 397, Supplementary figure 3B, Supplementary figure 4, Supplementary methods - Cell cultures, antibodies and reagents, Supplementary methods - Transfection of shRNA/siRNA and generation of stable lines with shRNA knockdown 2x).
Response: We have corrected the spelling errors.
3)"triple-negative" (line 19), "promoter" (lines 25, 203, 229, 233, 236, 257, 340, 341, and 348), "treatment" (lines 27 and 416 and Supplementary figure 3 2x), "kinase" (line 63), "invasion assay" (lines 81 and 93), "white" (line 83), "canonical" (line 94), "signaling" (line 103), "polycomb" (line 128), "not" (line 135), "poly" (line 174), "glycohydrolase" (line 174), "experiments" (line 197), "polyacrylamide" (line 216), "chromatin" (lines 231, 233, 237, 240, 241, 255, 342, Supplementary figure 5 2x, and Supplementary methods - Co-immunoprecipitation and chromatin immunoprecipitation), "green" (lines 266 and 282), "relapsed free survival" (line 299), "dataset" (line 309), "low" (line 317), "log-rank" (line 326), "transcriptome analysis" (line 376 and Supplementary methods - Whole transcriptome analysis), "colony" (line 385 and Supplementary methods - Cell proliferation and colony formation assay 2x), "using" (line 387), "chamber" (line 389 and Supplementary methods - Wound healing assay and Boyden chamber invasion assay), "chromatin immunoprecipitation" (line 399 and Supplementary methods - Co-immunoprecipitation and chromatin immunoprecipitation), "inflammatory and oncogenic" (line 414 and Supplementary figure 1), "gene" (line 415 and Supplementary figure 2), "chromatin and downregulated" (line 418 and Supplementary figure 5), "heatmap" (Supplementary figure 2), "in" (Supplementary figure 3), "chromosome" (Supplementary figure 4), "glutamine" (Supplementary methods - Cell cultures, antibodies and reagents), "pictures" (Supplementary methods - Cell proliferation and colony formation assay) should all be spelled with the first letter(s) lower-case.
Response: We have corrected the capitalization errors in the text.
4) Please replace "Co-Immunoprecipitation" with "Co-immunoprecipitation" (line 399 and Supplementary methods - Co-immunoprecipitation and chromatin immunoprecipitation 2x).
Response: We have corrected the capitalization in the text and Materials and Methods section.
5) The statement "to regulate transcription CCL2" (line 24) is not grammatically correct. Please replace with "to regulate CCL2 transcription" or "to regulate transcription of CCL2"
Response: We have replaced the statement with “to regulate CCL2 transcription”.
6) Please remove period from "upon CCL2. Treatment" (line 27).
Response: We have removed the period after “CCL2”.
7) "mechanism" could be in plural form "mechanisms" (line 46).
Response: We have corrected “mechanism” to “mechanisms”.
8) Please swap "MB-231" (line 73) and "MDA-MB-231 (MB-231)" (line 75) as the abbreviation for MDA-MB-231 cells should be mentioned at the place of its first occurrence (line 73).
Response: We have made the change to "MDA-MB-231 (MB-231)" in current line 73.
9) The label "(Figure 1B Left panel)" should read "(Figure 1B Right panel)" (line 79).
Response: We have made the change to “(Figure 1B, right panel)” now at line 81.
10) There is a discrepancy between PJ34 concentration used in the Boyden chamber invasion assay between the text (20 uM) (line 84) and Figure 1C (25 uM) (line 87).
Response: We have corrected the invasion assay value for PJ34 to 20 µM in the Figure legend 1C.
11) There is another problem in Figure 1A (line 87) as the number of samples probed (9) is different from the number shown as GAPDH loading controls (10). Please recrop the GAPDH image so that it better corresponds to what is shown in Figure S7.
Response: We have corrected the figure to show the correct number of lanes for the GAPDH control in Figure 1A.
12) The last issue about Figure 1A (line 87) is that there are two small unwanted marks around the square framing the PAR image part (on the outer edge of the upper right corner side just below the MB-231 and MB-468 labels). Although this is just a cosmetic detail, the western blot panel would look graphically more professional without these two redundant spots.
Response: We have removed the unwanted marks from Figure 1A.
13) Please equip one of the two micrographs in Figure 1B (line 87) with a scale bar.
Response: We have added a scale bar on the bottom panel of Figure 1B. The length of the scale bar is added in the legend.
14) Please equip one of the four micrographs in Figure 1C (left) (line 87) with a scale bar.
Response: We have added scale bars on Figure 1C. The scale bars are placed on the left panel on the bottom and the right panel on the top right pictures. The length of the scale bar is added in the legend.
15) Due to poor resolution and/or small image size it is very hard to distinguish cells present on the Boyden chamber invasion assay micrographs in Figure 1C right (line 87). Would it please be possible to increase the resolution or provide enlarged versions of these two (treated and untreated) images so that readers can see what is and what is not a cell. Alternatively, enlarged images can be included in the supplementary file.
Response: We have adjusted the contrast and added to the legend: “Dark objects on the pictures are cells. One arrow is shown in each panel pointing to one of the cells”. For Boyden chamber assay, cells were fixed with 4% paraformaldehyde and stained with 0.1% Crystal Violet, a protein binding dye. The cells show a dark blue color after staining without any cellular details visible under bright field. The picture are here converted to grayscale for better contrast. The cells in the pictures appear as solid dark streaks or dots. The enlarged picture is in supplementary figure 12 with the description above. We have also included the cell staining details in the method section.
16) Please equip one of the two micrographs in Figure 1C right (line 87) with a scale bar.
Response: We have added a scale bar on the right picture in Figure 1C with scale bar length mentioned in the legend.
17) Please indicate for how long were MB-231 cells treated with PJ34 inhibitor in the Boyden chamber invasion assay in Figure 1C right (line 87) and in the dose-response transcript level measurement in Figure 3A right (line 190).
Response: We have included in the legend of 1C as “Cells were incubated with PJ34 for the duration of the experiment” (current line 97). In figure legend 3A, we include the word “overnight” (Current line 202).
18) It is not clear in Figure 1D (line 87), which part represents PARP1-depleted BT549 cells and which part is for PJ-34-treated MB-231 cells. Please indicate this clearly and/or split the panel into two (Figure 1D and new Figure 1E). Organize figure legend accordingly.
Response: Figure 1D represents data form BT549 cells except for the chemokines shown on the bottom right. Those genes showed similar pattern of expression with either inhibition or knockdown. We have corrected and removed the PJ34 part from the legend in Figure 1D.
19) Please provide more detailed description in the legend to Figure 1D (line 87) about the meaning of the presented data:
- a) What does "Overlap" mean?
b) What does red and blue color coding indicate?
c) What does "Genes" column tell?
Response: We have added the text as follows in current line 101-106: “Overlap showing the % of differentially expressed genes that belong to the pathways shown. The color bar below indication downregulation (red) and Upregulation (blue) with PARP1 knockdown. “Genes” column Bottom left showing number of differentially expressed genes that belong to the molecular pathway category. Chemokines shown in (D) Bottom Right show similar patterns of expression with either PARP1 inhibition or knockdown.”
20) Please make cytokine P values fully visible in Figure 1D (line 87). Also, the colon ":" character in "Pathway:" in the upper left corner seems to be cut through.
Response: We have corrected this with a new figure showing the full p-vales in the panel 1D.
21) Although the authors claim that "PARP1 knockdown showed a significant effect on cytokine signaling" (line 101), "In the Ingenuity Pathway (IPA) analysis, interferon Signaling and IL-17A were significantly affected" (line 102), and "PARP1 inhibition modulated inflammation-related pathways" (line 105), from the entire paragraph is not clear what the trend was. Please make it clear in the text whether these pathways upregulated or downregulated following PARP1 inactivation?
Response: We have clarified in the text as follows: “Interferon pathways showed predicted activation with IL-17A did no present any pattern in IPA analysis. PARP1 knockdown similarly affected cellular processes involving antigen presentation and activation of IRF (Interferon Regulatory Factors). In these cases, overall change in the pathways remained neutral due to lack of published literature. Overall the data suggested that PARP1 can modulate inflammation-related pathways. Although, more work will be needed to understand the mechanisms involved”. (Current line113-118).
22) The sentence "Other routes, such as antigen presentation and activation of IRF, also suggested that PARP1 inhibition modulated inflammation-related pathways." (line 105) is rather confusing as antigen presentation pathway and activation of IRF by cytosolic pattern are in the shPARP1 column. Please reserve the word inhibition for chemical intervention only or indicate for which particular case was PJ34 treatment used in Figure 1D (line 87).
Response: We have modified it as: “PARP1 knockdown similarly affected cellular processes involving antigen presentation and activation of IRF (Interferon Regulatory Factors).” (Current lines 114-115).
23) Please replace "Ingenuity Pathway (IPA) analysis" with "Ingenuity Pathway Analysis (IPA)" (line 102).
Response: We have corrected it as "Ingenuity Pathway Analysis (IPA)" (Current line 111)
24) "Ingenuity Pathway Analysis (IPA)" (lines 106) and "Ingenuity Pathways (IPA) analysis" (line 272) could be shortened to "IPA".
Response: We have shortened “Ingenuity Pathway Analysis” to IPA on the current lines 114,119 and 285.
25) The authors state that "Ingenuity Pathway Analysis (IPA) showed that the significant pathways upregulated in MB-231 are the EIF2 signaling with 42% overlap, oxidative phosphorylation showing 107 49% overlap with the genes of those pathways Supplementary Figure 1). Protein translation-related EIF4 and p70S6K pathways were also positively regulated. Along with that, Sirtuin signaling showed significant changes." (line 106), however there is no mention of EIF2, oxidative phosphorylation (OXPHOS), EIF4, or p70S6K in Supplementary Figure 1. Please modify this figure to clearly contain all this factors, provide tabulated source data, or indicate as "(data not shown)".
Response: We have now included the PJ34 specific data in supplementary figure 1A top panel showing the details of pathways with PJ34 treatment as determined by IPA analysis. We have referenced the figure in the next.
26) The formulation of the sentence "A total of 1059 genes were differentially expressed after utilizing EDGR for differential gene expression" (line 114) could be improved as:
- a) The number of genes with altered expression following shRNA-mediated PARP1 depletion 1059 is different from 1095, which is indicated in Supplementary Figure 1.
- b) This sentence does not mention the fact that this number refers to shRNA-mediated PARP1 depletion, which is confusing because this is being compared to PJ34 treatment in the following sentence: "On the other hand, PJ34 treatment significantly affected the transcriptional status of nearly 5759 genes" (line 115).
Response: We have modified this as: “Next, we explored the overlap between PJ34 mediated gene signatures and differentially expressed genes in response to PARP1 knockdown. A total of 1059 genes were found to be differentially expressed in BT549 cells upon PARP1 knockdown as determined by the edgeR method. On the other hand, PJ34 treatment in MB-231 cells significantly affected the transcriptional status of nearly 5759 genes.” (Current line 125-129).
27) Please replace "Broad" with "Broad Institute" (page 121).
Response: We have changed to “Broad Institute Oncogenic Signature.” (Current line 133)
28) There is a difference in the name of the two genes (CXCL13 and ADMA8) noted in the text as the outcome of the leading edge analysis (line 127) and in Supplementary figure 2 (CXCL3 and ADAM8). Please fix.
Response: We have corrected to CXCL3 in the main text. (Current line 139).
29) Please split "survival movement" using a comma as these are two different processes (line 136).
Response: We have added a comma and modified it as “cell survival, movement” (Current line 46) and “survival, cellular movement”. (Current line 148).
30) Please correct "(Figure & D)" to "(Figure 2C,D) (line 148).
Response: We have corrected as “Figure 2C & D”. (Current line 160).
31) The y axes for graphs in Figure 2C & D (line 152) don't have exactly the same height, which makes comparison between MDA-MB-231 and BT-549 cells less accurate.
In addition, the font seems to be different between these two graphs as well. Please replot data using consistent size and text font.
Response: We did the experiments in the two lines, namely MB-231 and BT549.Our comparison is only between the values belonging to the same cell line with the siRNA treatment. We compared siRNA treatment with scrambled control in each cell line. We did not intend to compare between the two different cell lines in this case. We have modified the fonts.
32) The amount of secreted CCL2 1900pg/ml mentioned in the text (line 169) does not exactly correspond to what is shown in Figure 3B (line 187) as from the graph the mean value seems to be less than 1900pg/ml by visual inspection.
Response: We have corrected the value of CCL2 to 1562 pg/ml with the text “decreased from 4094 pg/ml to 1562 pg/ml (Figure 3B)”. (Current line 181).
33) The statement "We found an 80% to 90% reduction in CCL2 transcript with PJ34 treatment in those two cell lines" (line 172) seems to be incorrect in that MDA-MB-436 cells don't reach such profound reduction following PJ34 treatment when consulting Supplementary figure 3A. Please use increased precision in gauging values from plotted figures.
Response: We modified it to: “While CCL2 levels were decreased in both cell lines, we found an 80% to 90% reduction in CCL2 in the HCC1937 cells.” (Current lines 183-184).
34) Reference to Supplementary figure 3B is missing. Please incorporate result description of Supplementary figure 3B into the main text.
Response: We added the following sentence: “In the HCC1937 cells, PARP1 was also localized to the CCL2 promoter as seen by chromatin immunoprecipitation with PARP1 antibody (Supplementary Figure 3B).” (Current lines 185-186)
35) Please remove one of the two sentences: "Interestingly, treating MB-231 cells with PDD 17273 (PARGi) resulted in increased PARylation (Figure 3C) [13]. Expectedly, total PAR levels increased after treatment with PARGi (Figure 3C Left panel)" (line 176) as they both inform on the same effect of increased PAR levels (PARylation) following PARG inhibitor treatment and hence are mutually redundant.
Response: We have removed the sentence “Expectedly, total PAR levels increased after treatment with PARGi (Figure 3C Left panel)".
36) Please replace "PDD 17273" with "PDD00017273" (line 177).
Response: We have corrected it to “PDD00017273”. (Current line 190).
37) Please add the word "treatment" after "PARGi and PJ34" (line 183).
Response: We have added the word “treatment” which reads as “PARGi and PJ34 treatment resulted in an increased expression of CCL2 compared to PJ34 alone.” (Current line 196)
38) Please realign western blot images for PAR and PARP1/GAPDH in Figure 3A (line 187) so that there is a visible vertical gap between the untreated and PJ34-treated western blot results for all the target proteins probed.
Response: We have added the modified figure with the vertical gap in figure 3A.
39) Please remove the small black redundant "minus" mark, which is just above the (A) label in Figure 3A (line 187).
Response: We have removed the redundant “minus” mark for the figure.
40) Please indicate for how long were MB-231 cells treated with PARG inhibitor in Figures 3C and 3D (line 187) in their respective legends.
Response: We have added “overnight” in the legend to read “after overnight PARG inhibitor (PDD 00017273) treatment in MB-231cells.” (Current lines 205-206)
41) Remove "Right: Western bolt showing PARP1 and PAR levels." from the legend to Figure 3A (line 189).
Response: We have removed “Right” form the legend of figure 3A and changed to “Left: Western bolt showing PARP1 and PAR levels”. (Current line 201-202)
42) Remove "(left)" from the legend to Figure 3A (line 191).
Response: We have removed “left” from figure legend 3A and changed to “Right: Relative CCL2 transcript levels after overnight PJ34 treatment in MB-231with PARP1 inhibitor PJ34.” (Current lines 202-203).
43) Remove "is on the right" from the legend to Figure 3B (line 191) as there is only one panel.
Response: We have removed “is on the right” from figure legend 3B.
44) Change "after PARG inhibitor, PARP1 inhibitor, and in the combination" to "in the presence of PARG inhibitor, PARP1 inhibitor, and the combination" (line 195).
Response: We have changed to “in the presence of PARG inhibitor, PARP1 inhibitor, and the combination.” The legend reads as follows: “Relative expression of CCL2 mRNA in the presence of PARG inhibitor, PARP1 inhibitor, and the combination of the two at 5 µM and 10µM doses respectively.” (Current lines 207-208).
45) Please replace "immune precipitation" with "immunoprecipitation" (line 222).
Response: We have changed to “immunoprecipitation”. (Current line 233).
46) Please replace "(Supplementary Figure 4B)" with "(Supplementary Figure 5B)" (line 226).
Response: we have removed Supplementary Figure 5B since the original image is not available. We have also removed mentions of this figure in the text.
47) Reference to Supplementary figures 5C and 5D are missing. Please incorporate result description of Supplementary figures 5C and 5D into the main text.
Response: We have removed those two figures from our article. Since it was not initially described, we did not discuss these figures in the text and removed the two panels in supplementary figure 5.
48) Please replace "treatment P65, specific" with "treatment, P65-specific" (line 226).
Response: We have removed the sentence since this was in reference to the supplementary figure 5B, which has been removed.
49) "Since we observed that NFκB and PARP1 cooperation" is not grammatically correct (line 228). Please correct.
Response: We have modified the sentence to read “Since we observed that P65 and PARP1 interact in breast cancer cells”. (Current lines 237).
50) Please replace "(Supplementary Figure 4A)" with "(Supplementary Figure 5A)" (line 241).
Response: We have corrected “Supplementary Figure 4A” with “Supplementary Figure 5”. (Current line 247).
51) Could the authors briefly comment on in the text as to why there is a decrease observed in the p65 signal in the nuclear fraction of MB-231 cells following PJ34 treatment in Supplementary figure 5A but not in Figure 4C?
Response: For the co-immunoprecipitation experiment, we added nuclear export blocker leptomycin-B. We believe this could be the reason for P65 retention in the nuclear extract after PJ34 treatment.
52) Please indicate for how long were MB-231 cells treated with PJ34 inhibitor in the p65 and PARP1 antibody chromatin immunoprecipitation assay and what was the dose of PJ34 used with p65 antibody chromatin immunoprecipitation in Figure 4D (line 244).
Response: We have added “after overnight treatment with 10 µM PJ34” in the figure legend of 4D. (Current line 262).
53) Please state what do the two dashed arrows drawn along the NFκB-binding motif indicate in the legend to Figure 4D (line 244).
Response: These indicate the chromatin IP primers designed around the P65 binding site. We added in the legend of figure 4D: “Dotted arrows showing the approximate location of the primers”. (Current lines 263-264).
54) There seems to be discrepancy between the serine amino acid residue number targeted by the phospho-p65 antibody: The sentence "Interestingly, in the presence of the PARP1 inhibitor p65 serine 436 phosphorylation, associated with NFκB activation, was reduced significantly without affecting the total level of p65" (line 210) refers to Ser436 in contrast to the sentence "Western blot showing activation of NFκB pathway probed with an antibody against phosphorylated p65 S536" (line 247) which points to Ser536. Please remove any ambiguities.
Response: The correct serine is 536 for phosphorylation of P65, which is a mark for activation as a transcription factor. We have corrected 436 “P65 serine 536 phosphorylation”. (Current line 223).
55) Replace "streptavidin HPR conjugated" with "HRP-conjugated streptavidin" (line 252)
Response: We have replaced "streptavidin HPR conjugated" with "HRP-conjugated streptavidin”. (Current line 260)
56) Replace "showing" with "showed" (line 266).
Response: We have modified the sentence to “Immunofluorescence with PAR antibody shows increased PAR levels (green) in rhCCL2-treated MB-231 cells.” (Current lines 275-276).
57) Replace "treated" with "rhCCL2-treated" (line 266).
Response: We have added rhCCL2-treated to read as response to 56 above: “Immunofluorescence with PAR antibody shows increased PAR levels (green) in rhCCL2-treated MB-231 cells.” (Current lines 275-276).
58) Please incorporate a sentence commenting on the result relevant to the phospho-ERK1/2 western blot in Figure 5A left (line 278) into the main text.
Response: We added the following sentences: “Expectedly, phosphorylation of ERK1/2 was increased upon addition of rhCCL2. We utilized ERK1/2 phosphorylation as an indication of effective rhCCL2 treatment since it is known to activate the Map kinase pathway.” (Current lines 273-275).
59) Please rescale the four micrographs shown in Figure 5A right (line 278) so that fine morphological details can be discerned inside cells. Alternatively, provide enlarged images as part of the supplementary file.
Response: We have added modified pictures with scale bars in figure 5A. We also included the enlarged images in supplementary figure 8.
60) Please equip one of the four micrographs shown in Figure 5A right (line 278) with a scale bar.
Response: We have added modified pictures with scale bars in figure 5A.
61) Reference to Figure 5B is missing. Please incorporate result description of Figure 5B (line 278) into the main text.
Response: We added the following text: “We also observed downregulation in ERK1/ (P44/42) phosphorylation with PJ34 treatment indicating inactivation of the MAP Kinase pathway (Figure 5B)”. (Current lines 276-277).
62) Please add the original western blot image for GAPDH as part of Figure 5B (line 278) into the supplementary file among other western blot images.
Response: We have added the original GAPDH image in supplementary figure 10.
63) Please provide tabulated data underlying the transcriptome analysis presented in Figure 5C (line 278) as an additional supplementary file so that all genes names are visible to the reader.
Response: We have included an excel file with the details of differentially expressed genes that shows 1.5 log2 fold changes in rhCCL2 treatment, PJ34 treatment in MB-231 cells and shRNA mediated knockdown of PARP1 in BT549 cells. We have also included the data for the IPA analysis shown in Figure 5C and GSEA in supplementary figure 1 in different sheets of the excel files.
64) Please provide more detailed description in the legend to Figure 5C left (line 278) as to what does red and green color coding indicate.
Response: We have added “Red color indicates upregulation with green indicating downregulation of mRNA transcription.” An enlarged version of the zoomed heatmap is in the supplementary figure 8 with this information added to the legend. (Current line 299-300).
65) The first and last gene names present in the zoomed heat map record are cut through in Figure 5C left (line 278). Please correct.
Response: We have corrected this in Figure 5C. The enlarged image is also provide in the supplementary figure 8. We have included the supplementary figure number in the legend,
66) Please remove the small intelligible gene description (left) as well as the small intelligible heat map legend (bottom) from Figure 5C left (line 278).
Response: We have removed the gene names from the rows from the small heatmap in Figure 5C.
67) Please indicate for how long and using what dose were MB-231 cells treated with rhCCL2 and PJ34 inhibitor in Figures 5C left and right (line 278).
Response: We have added the information as “rhCCL2 (1 hrs treatment as in A) and PJ34 (10 µM Overnight treatment) treated groups” in Figure legend of 5C. (Current line 298).
68) Please make IPA pathway names fully visible in Figure 5C right (line 278).
Response: We have added a re-edited image for Figure 5C IPA data. An enlarged version in include in the supplementary figure 8.
69) "Activation z-score", "-1.203", and "0.533" descriptors are difficult to read because the font color is gray in Figure 5C right (line 278). Would it be possible to change the font color to black?
Response: We have added the new figure with better contrast. The values are changed to show better contrast in the heatmap color.
70) Please add length information for the scale bars displayed in Figure 5D left (line 278) to the respective legend. Also, please remove the small intelligible symbols from above all scale bars in the same figure.
Response: We have added the information about the scale bar in the legend as “Scale bar 100 µm.” We have also removed the small text above the scale bar line.
71) Graph "20 uM PJ34" and "20 uM PJ34 + rhCCL2" legends in Figure 5D right (line 278) are plotted using indifferent font colors. Please replot the respective graph legend using orange font color.
Response: We have corrected the figure to show proper color corresponding to graph line for the legends in figure 5D right in the wound healing image.
72) Please indicate for how long were MB-231 cells treated with rhCCL2 in Figure 5D (line 278).
Response: We have added the following text to the legend: “Near confluent cells were pretreated with PJ34 before rhCCL2 was added at the dose of 50ng/ml after wound creation. Treatment was continued for 12 hrs.” (Current line 303-305).
73) Please distinguish "Left:" and "Right:" sections for the western blot and immunofluorescence result, respectively, in the legend to Figure 5A (line 279).
Response: We have “Left” and “Right” in the legend of Figure 5A to read: “Left: Recombinant human CCL2 (rhCCL2) can increase PAR levels in MB-231cells. MB-231cells were serum-starved overnight and treated with 50ng/ml rhCCL2 for 1 hrs. Western blot showing total PAR level upon treatment. Right: Immunofluorescence images showing PAR (green) in cells after rhCCL2 treatment.” (Current lines 290-293)
74) Replace "rhCCL2 and PJ simultaneously on" with "simultaneous rhCCL2 and PJ34 treatment" (line 288).
Response: We have corrected to “simultaneous rhCCL2 and PJ34 treatment” in the legend of Figure 5D. (Current line 302).
75) The sentence "PPAR signaling and PARG alpha, ERK5 pathways are upregulated or significantly affected by rhCCL2 treatment" (line 291) is not fully correct as PARG alpha seems to be absent from the IPA list in Figure 5C right (line 278). Please include additional data into Figure 5C, provide tabulated source data, or indicate as "(data not shown)" for PARG alpha.
Response: We have modified the sentences as: “PPAR signaling and PPARα/RXRα activation are upregulated or significantly affected by rhCCL2 treatment. However, these pathways are down-regulated in PJ34 treatment along with ERK5 signaling.” (Current lines 306-308). We have removed “PARG alpha” from the text. The details of the IPA list is provide in the excel file in a separate sheet.
76) Replace "Cbioprtal" with "cBioPortal" (line 301).
Response: We have modified “Cbioportal” to “cBioPortal” in current line 315 as well as in the method section. (Current line 519).
77) Although the authors claim that "We found that CCL2 levels are significantly upregulated in Claudin-low and Basal type cancer in the METABRIC dataset of 2509 primary breast tumors with 548 matched standard samples" (line 301), only Claudin-low data set is indicated as significant in Figure 6A left (line 307). Correct the abovementioned sentence or indicate proper statistical significance in the respective figure using asterisk(s).
Response: We have corrected it and removed Basal, since it did not have significant p-value with respect to normal samples in Figure 6A. The current sentence is as follows with Basal removed and “standard” replaced with “normal”: “We found that CCL2 mRNA levels are significantly upregulated in Claudin-low type cancer in the METABRIC dataset of 2509 primary breast tumors with 548 matched normal samples.” (Current lines 315-317).
78) Similarly, correct the statement for p65: "However, P65 did not show any significant transcriptional upregulation except moderate upregulation in the Basal type" (line 304) or indicate statistical significance in Figure 6A right (line 307) using asterisk(s).
Response: We did not observe any statistically significant upregulation by ANOVA in the Basal even though the average value was higher. Hence, we did not include any p-value for this set.
79) In addition, indicate the meaning of numbers "199", "199", "220", "679", "461", "146" to the legend of Figure 6A left (line 307).
Response: We have included “Numbers in the figure represent total number of patients in each group.” In the figure legend 6A. (Current lines 325-326).
80) Could the authors please make it more clear in the text what is the difference between Figure 6B left and right (line 307)? Both figures refer to mRNA expression in the graph but only Figure 6B right but not left is described as such in the text: "Interestingly, CCL2 and P65 expression are positively correlated in the dataset" (line 319). Please indicate CCL2 and p65 expression as "mRNA expression" in this sentence, if mRNA expression is indeed being plotted in Figure 6B left. Please use reasonable amount of stringency when describing experimental data sets.
Response: We have mentioned these as mRNA expression in the legend of Figure 6. The data are from microarray showing mRNA expression for CCL2 and P65. Figure 6B right is showing the linear regression line while left is the correlation. Both of which are methods as indicator of whether two gene expressions are dependent on one another. We modified the line in the legend “Linear regression showing association between P65 and CCL2 mRNA expression.” (Current line 329).
81) Please indicate what are the two dotted lines in the legend to Figure 6B right (line 307).
Response: We included the following phrase in the legend of figure 6B: “Dotted lines showing confidence bands representing the confidence interval of the best fit line (solid).” (Current line 329-330).
82) Please replace "*p CCL2-P65 High with Low" with "*p CCL2-P65 High vs. Low" in Figure 6C (line 307).
Response: We have replace with "*p CCL2-P65 High with Low" with “*p CCL2-P65 High vs. CCL2-P65 Low” in the Figure 6C.
83) Please include definition for CCR2 in the legend to Figure 6D (line 307).
Response: We added the following lines in the legend of figure 6D: “CCR2 is shown as the cognate receptor for CCL2. CCL2 binds to CCR2 and activates MAP Kinases, which results in the downstream activation of PARP1 and P65 followed by recruitment at the CCL2 promoter.” (Current lines 334-336).
84) Swap "CCL2 (Right) and P65 (left)" with "CCL2 (left) and P65 (right)" (line 309).
Response: We have corrected the text in Figure 6A legend as “METABRIC Dataset showing CCL2 (Left) and P65 (Right).” (Current line 323).
85) Replace "PAM50-Clausdin Low" with "PAM50-Claudin low" (line 310).
Response: We have corrected “PAM50-Clausdin Low” with “PAM50-Claudin Low” in the figure 6A legend. (Current line 325).
86) The sentence "RELA/P65 expression is higher in basal (Turkeys multiple comparison. ** p < 0.001 with normal) and Claudin-low type" (line 312) is confusing because only Basal but not Claudin-low type are considered significant in the sentence "However, P65 did not show any significant transcriptional upregulation except moderate upregulation in the Basal type" (line 304) and in Figure 6A right (line 307). Please revise both sentences and if p65 Claudin-low data set is indeed significantly different from Normal, indicate this fact in the respective figure using asterisk(s).
Response: We rephrased and corrected this as: “CCL2 expression is significantly higher in the Claudin-low type (One way ANOVA followed by Tukey’s multiple comparison. ** p < 0.001 vs. Normal).” (Current lines 326-327). We removed the Basal part it was higher but not significant. We also removed P65 expression as it was not significant from the legend of figure 6.
87) Replace "basal" with "Basal" (line 312) and "with normal" with "vs. Normal" (line 312).
Response: We have removed “basal” and corrected to “vs. Normal”. (Current line 327).
88) Replace "Turkeys" with "Tukey's" (line 312 and Supplementary figure 6).
Response: We have corrected “Turkey’s” to “Tukey’s” in Figure 6 and Supplementary Figure 6.
89) Please indicate CCL2 and p65 expression as "mRNA expression" (2x) in the sentence "Left: Correlation of CCL2 and P65 in patients. Spearman correlation with a two-tailed test for P. Right: Linear regression between P65 expression and CCL2 expression" (line 313), if mRNA expression is indeed being plotted in Figures 6B left and right (line 307).
Response: We have added “mRNA expression” if Figure 6A and Figure 6B for correlation and regression. (Current lines 323, 328-329).
90) Replace "Kaplan Myer" with "Kaplan-Meier" (line 314).
Response: We have corrected to “Kaplan-Meier” in the legend of figure 6C. (Current lines 332-333)
91) Replace "kmplot" with "KMPLOT" (line 316).
Response: We have replaced “kmplot” with “KMPLOT” in Figure 6C. (Current line 331)
92) Please replace "r 0.16" with "r = 0.16" (line 320).
Response: We have replaced "r 0.16" with "r = 0.16". (Current line 338).
93) The classification formulated as "classify a patient as high expressing with the low expression for each of the genes" (line 323) is not clear. Please rephrase.
Response: We rephrased this as “A median split was used for CCL2 and P65 expression levels to stratify patients in high-expression or low-expression groups.” (Current lines 341-342).
94) Could the authors briefly comment in the text on the fact that the relapsed free survival was shorter for the CCL2 high group when compared to the CCL2-p65 group?
Response: we found that CCL2 high samples were not statistically significantly higher than CCL2-P65 high or just P65 high samples by ANOVA. This could be due to not having sufficient number of patients in the CCL2 only data set, as most patients were censored or relapsed (event) happened.
95) Replace "contribute" with "contributes" (line 326).
Response: We have corrected this. (Current line 344).
96) Could the authors please add a section about whether the interaction between PARP1 and p65 has been previously observed in other studies in the Discussion section?
Response: We have added the following text: “PARP1 is also known to directly interact with both P65 and P50 as a coactivator, which does not require enzymatic activity” with the reference “Hassa, P.O.; Covic, M.; Hasan, S.; Imhof, R.; Hottiger, M.O. The enzymatic and DNA binding activity of parp-1 are not required for nf-kappa b coactivator function. The Journal of biological chemistry 2001, 276, 45588-45597.” (Current lines 356-358).
97) Replace "senescence" with "senescent" (line 337)."
Response: We have corrected to “senescent”. (Current line 355).
98) Replace "examine" with "be examined" (line 349).
Response: We have corrected this to “be examined”. (Current line 371).
99) Replace "(Supplementary Figure 5)" with "(Supplementary Figure 6)" (line 356).
Response: We have modified this as along with change in the figure number: “In the TCGA breast cancer data, (UC Santa Cruz, xenbrowser.net, Nature 2012), two of the factors in this study (PARP1 and CCL2) showed significant overexpression in the Basal type of tumors with respect to the Luminal-A or B subtype (Supplementary Figure 6).” (Current lines 375-377).
100) Replace "cells sequenced" with "cells was sequenced" (line 377).
Response: we could not locate this possibly due to modifications made in the text already.
101) Remove the latter "in duplicate" from the sentence "ELISA was performed twice in duplicate using the Human CCL2/CCL2 Quantikine ELISA Kit 379 (R&D Biosystem DCP00) in duplicate" (line 379 and Supplementary methods - ELISA for human CCL2) and please change "CCL2/CCL2" to "CCL2/MCP-1" in the same sentence.
Response: We have corrected this in the method section by removing duplicated words from the ELISA method. It is currently reads as follws: “ELISA was performed twice in duplicate using the Human CCL2 Quantikine ELISA Kit (R&D Biosystem DCP00) following the manufacture’s instruction. CCL2 levels (pg/ml) were normalized to 1x10^6 cells.” (Current lines 420-422).
102) Please add "cells" after "BT-549" (line 397).
Response: We have added “cells” after BT549 and changed all “BT-549” to “BT549” for consistence thought the text.
103) Please remove "and" or fill in the missing word(s) from the sentence "PARP1 siRNA and was used to knockdown PARP1 in MB-231 and BT-549 cells" (line 397).
Response: We have removed “and”. We included in method section the following: “ siRNAs targeting human PARP1 (Thermo Fisher Scientific s1099 and s1097) and control pool were being used to knockdown PARP1 in MB-231 and BT549 cells”. (Current lines 470-471).
104) Please provide legend to Supplementary figure 1. Namely, is not clear:
- a) What is displayed on the x axis in Supplementary figure 1B?
b) What does red and blue color coding indicate?
c) What do the black vertical lines adjoining the red-blue color map mark?
d) What do the small red marks under the x axis of the plotted graph for IL2_UP.V1_UP in Supplementary figure 1B represent?
e) What is the meaning of and/or abbreviation definition for size, ES, and NES in Supplementary figure 1B?
Response: These are from GSEA program, Size is the number of genes in the dataset that match with our gene list ES and NES are enrichment score and normalized enrichment scores respectively. We have included a legend in supplementary Figure 1 to explain the figure which is as follows:
“(A) Top: Pathways affected by PJ34 treatment in MB-231 cells. Bottom: Venn diagram of differentially expressed genes in MB-231 cells after PARP1 inhibition with PJ34 and shRNA mediated knockdown in BT549 cells. (B) Gene Set Enrichment Analysis (GSEA) on the 656 overlapping genes from (A) between PARP1 inhibition and knockdown is shown. Analysis was performed on the MsigDB oncogenic signature (c6.all.v6.2.symbols.gmt). Name: Gene expression set name from MsigDB. Size: number of genes belonging to the particular gene set from the list of 656 genes. ES= Enrichment Scores (in green line) (Number showing the degree to which the gene is overrepresented in the dataset)., NES=Normalized Enrichment Score (Actual ES/mean(ES against all permutation of the dataset), middle lines shows member of the gene sets found in the gene list provided. The red to blue bar along x-axis denotes positive (red) to negative (blue) enrichment. The lines also represents leading edge subsets. From: https://www.gsea-msigdb.org/.”
We have also include the IPA analysis of canonical pathways for PJ34 treatment in the supplementary figure 1A.
105) Please provide color legend for the heatmap values in Supplementary figure 2.
Response: GSEA program (v3) did not provide any color scale for the leading edge analysis directly however darker color represents higher enrichment score.
106) The statement "The gene sets on the shows enrichment" is not grammatically correct in the legend to Supplementary figure 2.
Response: In supplementary figure 2, we have the modified the text as “The leading edge subset of genes is shown after enrichment as found in the GSEA analysis using pre-ranked method. Clustering of genes was removed for clarity. Heatmap is showing log2 fold expression rank values of the genes listed on the column. Genes set are shown on the rows of the heatmap. The colors on the heatmap shows range of expression values with darker color representing higher log2 fold expression difference.”
107) Please replace "genes listed on the column" with "genes listed in the column" in the legend to Supplementary figure 2.
Response: Please see response to 106. We rewrote the legend/description of the supplementary figure 2.
108) Please add error bars to Supplementary figure 3B and indicate if any treatments were made to HCC1937 cell prior to PARP1 immunoprecipitation in the respective legend.
Response: We have added the error bars and added in the figure the following text: “Chromatin IP with PAPR1 antibody In BRCA1 negative HCC1937 at the CCL2 promoter under basal condition without any treatment.”
109) Please remove the small unnecessary "minus" mark in Supplementary figure 4, which is on the right side above the purple MCF7 dataset.
Response: We have corrected it as the top CCL2 gene locus picture had the line at the bottom, which was showing up. We have edited the picture to remove the line.
110) The label "shPARP1" is cut through in Supplementary figure 4. Please fix.
Response: We have moved the shPARP1 label to show it in full.
111) Please change "cell" and "Cell" to "cells" in the legend to Supplementary figure 4 and in Supplementary methods - Co-immunoprecipitation and chromatin immunoprecipitation.
Response: We have made the corrections to cells in supplementary figure 4 and methods in “Co-immunoprecipitation and chromatin immunoprecipitation”.
112) Please provide original western blot images for all targets probed in Supplementary figures 5A–D into the supplementary file among other western blot images.
Response: We have modified the figure and removed 5C and 5D, since they were not discussed. We have removed 5B as the original image is not available.
113) Please indicate for how long were MB-231 cells treated with PJ34 inhibitor for each of Supplementary figures 5A–D.
Response: We have removed supplementary figure 5B,C and D. 5A has been replaced with a new western figure with following text “Cytoplasmic (Cyto), nuclear (Nuc), distribution of PARP1 and P65 with or without overnight PJ34 treatment.”
114) It is not clear why IKKα (and not IKKα/β) served as the total signal for comparison in Supplementary figure 5D (when phospho-IKKα/β was used)?
Response: We have removed the figures since this was not mentioned in the text. For all blots, phospho antibodies are carried out first. We follow that with stripping, washing with TBS and incubation with BSA. We then incubate with total protein antibodies.
115) Please indicate statistical significance in Supplementary figure 6 using asterisk(s).
Response: We have included the significant p-values in supplementary figure 6 for PARP1 and CCL2 with ANOVA followed by Tukey’s multiple comparison, which corresponds to Basal-like with LuminalA/B.
116) Please comment on briefly as to the definition of "Basal-Like" tumor types in the legend to Supplementary Figure 6.
Response: This figure was generated using data from the “xenbrowser.net TCGA Breast Cancer (BRCA) data set. It is mentioned as basal-like. This group should be same as METABRIC basal type data.
117) Please label the panel demonstrating CCL2 expression as "PAM50 Subtype" in Supplementary figure 6 for increased consistency with other panels.
Response: We have corrected it from “PAM50” to “PAM50 subtype”.
118) Replace "All Cases" with "In all cases," in the legend to Supplementary figure 6.
Response: We have modified the legend in supplementary figure 6 and removed it.
119) Please provide a brief comment on the meaning of the row of numbers (e.g. 8.5, 9.4, 5.7), in both Supplementary figure 7 and 8, that appear along some of the western blot images.
Response: We removed those intensity values from the western blot pictures as those not used in the paper.
120) Please relabel "Western blots of Figure 3D" to "Western blots of Figure 3C" in Supplementary figure 7.
Response: We have correctly labelled as “Western blots of Figure 3C”. Supplementary figure number is now “9” from “7” due to addition of more supplementary images.
121) Please mark the lower set of bands as GAPDH using a red arrow in the PARP1 western blot belonging to Figure 3C in Supplementary figure 7. In addition, the PARP1 label seems to be cut through in the same western blot image. Please fit the label in.
Response: We have added the GAPDH using red arrow in supplementary figure 9 (previously supplementary figure 7). We have also made all labels including PARP1 clearly visible.
122) Please relabel "Western blot of Figure 4C" to "EMSA result of Figure 4B" and "Western blots of Figure 4D" to "Western blot of Figure 4C" in Supplementary figure 8.
Response: We have corrected "Western blot of Figure 4C" to "EMSA result of Figure 4B" and "Western blots of Figure 4D" to "Western blot of Figure 4C" in Supplementary figure 10 (previously supplementary figure 8).
123) Please provide catalog number for the DMEM/F12 medium used for cell cultures in Supplementary methods - Cell cultures, antibodies and reagents.
Response: We have included the catalogue no. of DMEM/F12 as “Thermo Fisher Scientific, Cat. No. 10565018.”
124) Please provide culture conditions for the following cell lines used: HCC1143, HS578T, HCC70, MB-453, MB-468 in the main text (line 374) and in Supplementary methods - Cell cultures, antibodies and reagents.
Response: We have included DMEM/F12 as the culture media. We cultured all cell in the same media with 10% FBS and antibiotics.
125) Please indicate antibody including its catalog number used for PAR, GAPDH, phospho-p65, p65 (total), phospho-p44/42, p44/42 (total), N-cadherin, vimentin, phospho-IKKα/β, and IKKα (total) in Supplementary methods - Cell cultures, antibodies and reagents.
Response: we have included the catalogue numbers of PAR, GAPDH and phospho-p44/42, p44/42 (total). The phosphor p65, p65 total antibodies are from the NFκB Sampler kit containing phospho-NF-κB p65 (Ser536) and total NF-κB p65 (CST Cat. No. 9936T). This information is added to the cells, antibodies and reagent section. We have removed “N-cadherin, vimentin, phospho-IKKα/β, and IKKα (total)” since the images were removed and no reference to those proteins are being made.
126) Please replace "Sample" with "Sampler" in Supplementary methods - Cell cultures, antibodies and reagents.
Response: We have corrected to “sampler” in the NFkB sampler kit.
127) Please add PDD 00017273, BAY 11-7082, TNFα, rhCCL2 and their manufacturer information into Supplementary methods - Cell cultures, antibodies and reagents.
Response: We have added the catalogue information in the “Cell cultures, antibodies, and reagents” section. (Current lines 405-411).
128) Please remove "Veliparib (ABT-888, Selleckchem S1004)" from Supplementary methods - Cell cultures, antibodies and reagents as this chemical was not used throughout the manuscript.
Response: We have removed “Veliparib” from “Cell cultures, antibodies, and reagents”.
129) Please change "RNA from treated or control cells were" to "RNA from treated or control cells was" in Supplementary methods - Whole transcriptome analysis.
Response: We changed “RNA from treated or control cells were” to “RNAs form treated and Control cells”. (Current line 413).
130) Please change "mirNeasy" to "miRNeasy" in Supplementary methods - Whole transcriptome analysis
Response: We have corrected "mirNeasy" to "miRNeasy" in the Whole transcriptome analysis section of materials and method.
131) "instruction" could be in its plural form "instructions" in Supplementary methods - RNA isolation and quantitative real-time PCR and in Supplementary methods - Gel shift Assay with NFκB probes.
Response: we have changed to the plural form "instructions" in “RNA isolation and quantitative real-time PCR” and in “Gel shift Assay with NFκB probes”.
132) Please replace "CFX96 touch thermal cycler" with "CFX96 Touch thermal cycler" in Supplementary methods - RNA isolation and quantitative real-time PCR.
Response: We have corrected to “CFX96 Touch thermal cycler” in the “RNA isolation and quantitative real-time PCR” section.
133) Please indicate "Cell proliferation and colony formation assay:" in Supplementary methods in bold font.
Response: Subheadings indicating different sections are not bold in the materials method section of the main text. We did not change this. All methods are now in the main text.
134) Please change "and assessed for growth inhibition after 72 hrs" to "and assessed for growth inhibition after 48–72 hrs" in Supplementary methods - Cell proliferation and colony formation assay.
Response: We have changed to “assessed for growth inhibition after 48-72 hrs.”
135) There seems to be a discrepancy between the amount of cells seeded between the statement "For Colony formation, 5000 cells were seeded in low attached 24 well plates in triplicates with or without PJ34 treatment" (Supplementary methods - Cell proliferation and colony formation assay) and the main text "For colony formation assay, about 1000 cells were seeded into low attachment 48 well plates for 7days" (line 387). Please fix and indicate for how long the cells were seeded for also in the Supplementary methods section statement.
Response: We have corrected the numbers to 1000 cells/well for colony formation assay in the method and result sections.
136) There is also discrepancy between the time used for measurement in the wound healing assay since the statements "Wound healing assay was performed on 96-well plate and wound was created using Biotek Autoscratcher and wound area was imaged using Gen5 software on Cytation 1 imager for every hour for 16 hours after treatment mentioned" (Supplementary methods - Wound healing assay and Boyden chamber invasion assay) and "The wound was created and imaged for every hour for 10-16 hours after the treatment mentioned" don't exactly match (line 390).
Response: We have corrected the discrepancy to 10-12 hours for the figures shown in 1 and 5. We have made the correction to “Wound healing and Boyden chamber invasion assay” section.
137) Please add the following instruction "The number of cells migrated onto the membrane was measured after overnight incubation." (line 392) also to the Supplementary methods section - Wound healing assay and Boyden chamber invasion assay.
Response: We have added to the materials method section. We have also added details about the staining procedure for Boyden chamber assay in the section “Wound healing and Boyden chamber invasion assay”. (Current line 449-452).
138) Please indicate how different fractions were prepared for Supplementary figure 5A in Supplementary methods section?
Response: We have added information about nuclear/cytoplasmic fraction preparation in “Gel shift assay with NFκB probes”. We used this for both EMSA (Figure 4B) and supplementary figure 5.
139) Please indicate the method of how successive western blot targets, such as the GAPDH loading control, were processed in Supplementary methods - Western blotting. For example, were western blot membranes stripped before reprobing? If yes, how?
Response: We have stripped and reused the same blot. We have added the information in the method sections. We have used “Restore™ Western Blot Stripping Buffer, Thermo Fisher Scientific, Cat. No. 21059”. We have added “All phospho protein antibodies were done first before stripping and probing with total protein antibodies and GAPDH loading control.” (Current lines 458-462)
140) Please replace "Cell Signaling technologies" with "Cell Signaling Technology" and "ThermoScientific" with "Thermo Fisher Scientific" in Supplementary methods - Western blotting.
Response: We have changed all instances of “Cell Signaling technologies" with "Cell Signaling Technology" and "ThermoScientific" with "Thermo Fisher Scientific" in the Materials and Methods section.
141) Please provide manufacturer for Optimem in Supplementary methods - Western blotting.
Response: We have added the relevant catalogue no. as “Thermo Fisher Scientific Cat no. 11058021”. (Current line 465).
142) Please change "Thermo Fisher Scientific" and "Thermo Fisher Scientific" to "Thermo Fisher Scientific" in Supplementary methods - Western blotting and in Supplementary methods - Co-immunoprecipitation and chromatin immunoprecipitation.
Response: We have changed all instances of "ThermoScientific" with "Thermo Fisher Scientific" in the Materials and Methods section.
143) Please change also "manufacture’s" to "manufacturer’s" in Supplementary methods - Western blotting.
Response: We have corrected "manufacture’s" to "manufacturer’s" in Supplementary methods - Western blotting.
144) Please describe the procedure for generating stable lines in Supplementary methods - Transfection of shRNA/siRNA and generation of stable lines with shRNA knockdown.
Response: We have provided the information regarding stable line selection with puromycin in the “Transfection of shRNA/siRNA and generation of stable lines with shRNA knockdown” section. (Current lines 473-475)
145) Please replace "17010086" with "17-10086" as the correct catalog number for the EZ-Magna ChIP A/G chromatin immunoprecipitation kit in the main text (line 400) and Supplementary methods - Co-immunoprecipitation and chromatin immunoprecipitation.
Response: We have replace the cat. No. with "17-10086" in the “. Co-immunoprecipitation and chromatin immunoprecipitation” section.
146) Please change "Cell Signaling" to "Cell Signaling Technology" in Supplementary methods - Co-immunoprecipitation and chromatin immunoprecipitation.
Response: we have replaced "Cell Signaling" to "Cell Signaling Technology" for all occurrences.
147) Please replace "manufactures" with "manufacturer’s" in Supplementary methods - Co-immunoprecipitation and chromatin immunoprecipitation.
Response: We have corrected "manufacture’s" to "manufacturer’s" in methods - Co-immunoprecipitation and chromatin immunoprecipitation.
148) Please change "Sonication of Cell were" to "Sonication of cells was" in Supplementary methods - Co-immunoprecipitation and chromatin immunoprecipitation.
Response: We have corrected "Sonication of Cell were" to "Sonication of cells was" in Method Co-immunoprecipitation and chromatin immunoprecipitation.
149) Please replace "output with 30S" with "output with 30s" in Supplementary methods - Co-immunoprecipitation and chromatin immunoprecipitation.
Response: We have corrected "output with 30S" with "output with 30s" in the methods - Co-immunoprecipitation and chromatin immunoprecipitation.
150) Please change "Following primers for CCL2 was" to "The following primers for CCL2 were" in Supplementary methods - Co-immunoprecipitation and chromatin immunoprecipitation
Response: We have corrected "Following primers for CCL2 was" to “The following primers for CCL2 were used flanking an NFκB binding motif” in Materials and Methods - Co-immunoprecipitation and chromatin immunoprecipitation section.
151) Please replace "standard deviation with statistically" with "standard deviation with statistical" in Supplementary methods - Statistics and Analysis of publicly available data.
Response: We have made the correction to “standard deviation with statistical significance” in Materials and Methods - Statistics and Analysis of publicly available data section.
152) Please change the web address "www.cbiopotal.org" to "www.cbioportal.org" in Supplementary methods - Statistics and Analysis of publicly available data.
Response: We have made the correction in the web address of cBioPortal to “cbioportal.org in Materials and Methods - Statistics and Analysis of publicly available data section.
153) Please describe methodology behind the immunofluorescence experiment in Figure 5A right (line 278) in the Supplementary methods section.
Response: We have added this to the Materials and Methods section 4.11 in the Materials and Methods with subheading “Immunofluorescence with MB-231 cells”.

Reviewer 2 Report
The manuscript by Dutta et al. is interesting and bring additional proof that CCL2 and PARP1 are important in breast cancer and are relevant targets. Using different cell line and methods to validate the involvement of both proteins is a strength of the manuscript. In addition to their own experiments, the authors use existing data to support their observations. Beyond cancer biology, Dutta et al provides new clues in the understanding of PARP1 functions.
However interesting, the experimental design could be improved and, for the results to support the conclusion, the statistics need to be verified.
Major comments:
Not using adapted nomenclature is confusing in the text, genes should be italicized.
Could the reduced proliferation be due to PJ34 toxicity? A cell death assay would add to the present work.
If PJ34 was not dissolved in cell culture media a vehicle control rather than no treatment should be added to the experiments.
The author used different cell line between PJ34 and shRNA experiments, this should be discussed.
Having at least 3 independent experiment would strengthen the results. The use of a student t test might not be the most accurate test to compare results of only 2 independent experiment. The author might consider asking a statistician advice to select the most appropriate test.
Shou et al. (PMC6629106) previously described an interaction between PARP1 and CCL2 expression this article should be referenced and discussed.
Minor comments:
Words shouldn’t randomly be capitalized, legend should be consistent (eg.: kd or KD, uM or µM)
Line 27 “ upon CCL2. Treatment.” => should the sentence stop after CCL2 ?
Line 57 “TNBC [3,4].TNBCs” => space missing
Line 79 and 85 “ (Figure 1B Left panel).” => add a comma after 1B to be consistent with the rest of your text
Line 100-101 “Isolated RNA was subjected to whole transcriptome sequencing on the Illumina platform.” => this seems more suited for the method section
Line 106 and 272 “Ingenuity Pathway Analysis” => no need to repeat this as IPA was define above.
Line 115 EDGR reference should be included
Line 118 no need to specify Venn diagram.
Line 120-121 “the Broad Oncology Signature (MsigDB Oncogenic Signature) [11]” => this seems more suited for the method section
Line 129: PARP1
Line 121: “, Log2 Fold Change” => this is part of the figure legend
Line 133: “CCL2 and CCL3 were”
Line 134 “red bars” => this is part of the figure legend
Line 136: What is “survival movement”?
Line 137 “IPA analysis” => this is part of the figure legend
Lines 167-168: “The CCL2 protein levels (pg/ml) were normalized to 10^6 cells” => this is part of the figure legend or method
Line 180: PARGi
Lines 181-182: “To this end, we subjected MB-231 cells to co-treatment with PJ34 and PARGi. Cells were pretreated with 10 µM PJ34 overnight, followed by PARGi at a 5 µM dose.” => co-treatment or pre-treatment?
Line 199: “Rel-A subunit” => precise it can be named P65 as it is the name used later in the manuscript.
Lines 213-216: “To this end, we employed EMSA with biotinylated probes for P65. We utilized the nuclear lysate from PJ34 treated or untreated control cells and incubated with p65 probes. After one hr., incubation, reaction mixture was separated on Polyacrylamide gel and transferred on to a nitrocellulose membrane. Using chemiluminescence,” => this should be in the method section
Lines 231-233: “We then harvested formaldehyde-fixed Chromatin from the cells and subjected to immunoprecipitation with P65 antibody. Precipitated DNA was analyzed by qPCR with primers flanking the NFκB consensus motif in the CCL2 Promoter. NFκB Chromatin IP data reflects % input in Figure 4D.” => this should be in the method section
Line 266: “Immunofluorescence images showing increased PAR levels (Green) in treated MB-231 cells.” => this should be in the legend or method section
Line 272 “Heat map dotted rectangle” => should be in figure legend not in the text
Line 297 “The graph on the right shows % wound closure with respect to time” => should be in figure legend not in the text
Line 354 “the Nature TCGA 2012 publication” => please improve the wording or just name the dataset
Figure 1: could you indicate the type of breast cancer next to the cell line for clarity.
Through the paper gene should be italicized
Could you align the panel labels, add scale bars and maybe adjust the size of the image in the figures
Figure 1D: Pvalues are duplicated
Figure S2: please add the legend for the heatmap
The Cbioprtal portal appears to have tumor data, thus CCL2 and P65 signaling may come from infiltrating immune cells, this should be discussed.
Author Response
Reviewer-2 Comments:
Comments and Suggestions for Authors:
The manuscript by Dutta et al. is interesting and bring additional proof that CCL2 and PARP1 are important in breast cancer and are relevant targets. Using different cell line and methods to validate the involvement of both proteins is a strength of the manuscript. In addition to their own experiments, the authors use existing data to support their observations. Beyond cancer biology, Dutta et al provides new clues in the understanding of PARP1 functions.
However interesting, the experimental design could be improved and, for the results to support the conclusion, the statistics need to be verified.
Response: We are thankful to the reviewer for kind comments regarding our manuscript. We have addressed all comments in the revised manuscripts with changes in the text, figures with updated references.
Major comments:
- Not using adapted nomenclature is confusing in the text, genes should be italicized.
Response: We have italicized for genes in the main text. For examples, CCL2 transcription has been changed to CCL2 transcription. We have also made the necessary changes in the figures as well.
- Could the reduced proliferation be due to PJ34 toxicity? A cell death assay would add to the present work.
Response: We have added a supplementary figure 7 with Annexin V/7AAD apoptosis detection in MB-231 cells upon overnight PJ34 treat. We added the text in result section as “This could possibly be attributed to cell proliferation defect as overnight treatment with PJ34 did not induce any significant apoptosis”. Details of the protocol was added to the method section.
- If PJ34 was not dissolved in cell culture media a vehicle control rather than no treatment should be added to the experiments.
Clarified that untreated is vehicle control (Water/DMSO).
Response: We have used PJ34 (PJ34 hydrochloride) dissolve in water as well DMSO. However, we have incorrectly used “untreated” to mean vehicle control. This has been clarified this in the text as “untreated vehicle control (hereafter “untreated”)”. (Current line 81).
- The author used different cell line between PJ34 and shRNA experiments, this should be discussed.
Response: Initially, we did the sequencing experiment in MB-231 with PJ34 and BT549 with shRNA. Since then, we have verified that CCL2 transcription is affected by PARP1 in both cell line with siRNAs. We also included PJ34 inhibition in HCC1937 and MB436 (Supplementary Figure 3). For CCL2, we have validated the effect of PARP1 in multiple cell lines to corroborate the findings from two different cell lines.
- Having at least 3 independent experiment would strengthen the results. The use of a student t test might not be the most accurate test to compare results of only 2 independent experiment. The author might consider asking a statistician advice to select the most appropriate test.
Response: It is likely that 3 experiments will be better in this case. We have consistently seen significant effect with our experiments with inhibitors, siRNAs in multiple cell line to provide support for CCL2 expressions and regulation. We will, however, be mindful of the suggestion in future. For statistical analysis, we understand that it might not be correct to assume that we have equal variance or normal distribution for the Cq values in RTPCR. Therefore nonparametric tests or ANOVA will be better. We carried out ANOVA for the RTPCR results and compared to the untreated samples.
- Shou et al. (PMC6629106) previously described an interaction between PARP1 and CCL2 expression this article should be referenced and discussed.
INCLUDE THE Reference
Response: We have included the reference and have modified the 2nd paragraph in discussion to show “Previously, PARP1 was shown to regulate CCL2 transcription in NK2 cells. We now show that PARP1 might be functional in multiple ways to regulate chemokines like CCL2 in breast cancer as well.” (Current line 359-361).
Minor comments:
- Words shouldn’t randomly be capitalized, legend should be consistent (eg.: kd or KD, uM or µM)
Response: We have corrected “kd/KD” to “kD” and have changed all “uM” to “µM”.
- Line 27 “upon CCL2. Treatment.” => should the sentence stop after CCL2?
Response: We have corrected this and removed the “period” after CCL2.
- Line 57 “TNBC [3,4].TNBCs” => space missing
Response: We have corrected this and added the space after TNBC. (Current line 57).
- Line 79 and 85 “(Figure 1B Left panel).” => add a comma after 1B to be consistent with the rest of your text
Response: We have added a “comma” in the text for consistency. We corrected the text as “Figure 1B, right panel”.
- Line 100-101 “Isolated RNA was subjected to whole transcriptome sequencing on the Illumina platform.” => this seems more suited for the method section
Response: We have removed the “illumine platform to read “Isolated RNA was subjected to whole transcriptome sequencing”.
- Line 106 and 272 “Ingenuity Pathway Analysis” => no need to repeat this as IPA was define above.
Response: We have corrected to the abbreviation
- Line 115 EDGR reference should be included
Response: We have included the reference with correction to the name as “edgeR”. We added the following reference “Robinson, M.D.; McCarthy, D.J.; Smyth, G.K. Edger: A bioconductor package for differential expression analysis of digital gene expression data. Bioinformatics 2010, 26, 139-140.”
- Line 118 no need to specify Venn diagram.
Response: We have remove “Venn Diagram” from the line.
- Line 120-121 “the Broad Oncology Signature (MsigDB Oncogenic Signature) [11]” => this seems more suited for the method section
Response: We have modified the sentence to “We subjected these common genes to Gene Set Enrichment Analysis (GSEA), primarily using the Broad Institute Oncogenic Signature set of genes.” and moved “MsigDB oncogenic signature” to Method. (Current lines 132-133).
- Line 129: PARP1
Response: We have changed PARP to PARP1.
- Line 121: “, Log2 Fold Change” => this is part of the figure legend
Response: We have removed this form the main text and moved to legend of figure 1D.
- Line 133: “CCL2 and CCL3 were”
Response: we have corrected “CCL2, CCL3 was” to ““CCL2 and CCL3 were”.
- Line 134 “red bars” => this is part of the figure legend
Response: We have removed the “red bar”.
- Line 136: What is “survival movement”?
Response: We have corrected above. We put a comma between survival and movement and in other place mention this as cellular movement. We have removed from the main text. It is now in legend and materials and methods section.
- Line 137 “IPA analysis” => this is part of the figure legend
Response: We have removed the “IPA analysis” from the parenthesis (Figure 1D).
- Lines 167-168: “The CCL2 protein levels (pg/ml) were normalized to 10^6 cells” => this is part of the figure legend or method.
Response: We have moved this line to figure legend 3B as well as Materials and Methods.
- Line 180: PARGi
Response: We have corrected “PARG treatment” to “PARGi treatment”.
- Lines 181-182: “To this end, we subjected MB-231 cells to co-treatment with PJ34 and PARGi. Cells were pretreated with 10 µM PJ34 overnight, followed by PARGi at a 5 µM dose.” => co-treatment or pre-treatment?
Response: In this experiments, cell were pretreated with PJ34. We changed this to “pretreatment” instead of “co-treatment”. We modified the text to “To this end, we subjected MB-231 cells to treatment with both PJ34 and PARGi. Cells were pretreated with 10 µM PJ34 overnight, followed by PARGi at a 5 µM dose.”
- Line 199: “Rel-A subunit” => precise it can be named P65 as it is the name used later in the manuscript.
Response: We have included “hereafter P65”” in parenthesis in current line 212. We have used P65 as the name after that.
- Lines 213-216: “To this end, we employed EMSA with biotinylated probes for P65. We utilized the nuclear lysate from PJ34 treated or untreated control cells and incubated with p65 probes. After one hr., incubation, reaction mixture was separated on Polyacrylamide gel and transferred on to a nitrocellulose membrane. Using chemiluminescence,” => this should be in the method section
Response: We have remove it from the main text result section to the materials and methods section under “Gel shift assay with NFκB probes”.
- Lines 231-233: “We then harvested formaldehyde-fixed Chromatin from the cells and subjected to immunoprecipitation with P65 antibody. Precipitated DNA was analyzed by qPCR with primers flanking the NFκB consensus motif in the CCL2 Promoter. NFκB Chromatin IP data reflects % input in Figure 4D.” => this should be in the method section
Response: We have moved it into the method section. We changed the following sentence to “. Expectedly, we observed increased binding of P65 after 1 hr. of TNFα treatment by Chromatin IP.”
- Line 266: “Immunofluorescence images showing increased PAR levels (Green) in treated MB-231 cells.” => this should be in the legend or method section
Response: We have changed this to “Along with that, Immunofluorescence with PAR antibody shows increased PAR levels (green) in treated MB-231 cells (Figure 5A, right).”
- Line 272 “Heat map dotted rectangle” => should be in figure legend not in the text
Response: We have removed this form the main text
- Line 297 “The graph on the right shows % wound closure with respect to time” => should be in figure legend not in the text
Response: We have moved this sentence into the legend of Figure 5D.
- Line 354 “the Nature TCGA 2012 publication” => please improve the wording or just name the dataset
Response: We have changed this to “TCGA breast cancer data, (UC Santa Cruz, xenbrowser.net, Nature 1012),”
- Figure 1: could you indicate the type of breast cancer next to the cell line for clarity.
Response: We have indicated that in the figure 1A showing the TNBC lines on the left and ER/PR+ lines on the right side of the blot.
- Through the paper gene should be italicized
Response: As stated previously, We have italicized for genes in the main text. For examples, CCL2 transcription has been changed to CCL2 transcription. We have also made the necessary changes in the figures as well.
- Could you align the panel labels, add scale bars and maybe adjust the size of the image in the figures
Response: We have corrected the figures with scale bars. Magnified view of Figure 1C Boyden chamber assay is included in the supplementary figure 12.
- Figure 1D: Pvalues are duplicated
Response: We have corrected the figure with p-values clearly showing without duplication.
- Figure S2: please add the legend for the heatmap
Response: We have added the following text “The leading edge subset of genes is shown after enrichment as found in the GSEA analysis using pre-ranked method. Clustering of genes was removed for clarity. Heatmap is showing log2 fold expression rank values of the genes listed on the column. Genes set are shown on the rows of the heatmap. The colors on the heatmap shows range of expression values with darker color representing higher log2 fold expression difference.”
- The Cbioprtal portal appears to have tumor data, thus CCL2 and P65 signaling may come from infiltrating immune cells, this should be discussed.
Response: This is an important point. cBioPortal data is from tumor core biopsies which might include significant amount of infiltrating immune cells such as macrophages or T-cells. It is difficult to tell with certainty whether gene expression profiles are solely due to contribution form only tumor cells. Therefore, we have included the following sentence in the discussion section: “Since, CCL2 or P65 can be upregulated in infiltrating immune cells, we cannot rule out contribution from the tumor microenvironment instead of only tumor cells.” (Current lines 378-379)

Reviewer 3 Report
This is an interesting report. It shows that PARP1 regulates CCL2 transcription through interaction with NFκB P65 subunit. In turn, CCL2 can also positively modulate PARP1 activity. The crosstalk between PARP1 and CCL2 regulates invasiveness in breast cancer.
The following issues need to be addressed to clarify and strengthen the manuscript:
- Other cytokines, such as CCL3, are also down-regulated under both PARP inhibition and knock down, which means PARP-1 could regulate the transcription of other cytokines. Please explain why you only focus on CCL-2 regulation?
- In the introduction section, there are no relevant citations describing PJ34. Please provide.
- In Fig. 1A, please specify which cell lines are triple negative cell lines.
- Please double check the statistical analysis used in Fig. 2, 3 & 4. In these figures, there are 1 control group and 2-3 independent treatment groups. Running two t-tests on the same data will increase type 1 error. Therefore, one-way ANOVA followed by a post-hoc analysis should be used.
- In line 334 and 335, “transcription of CCL2 and CCL2” should be “transcription of CCL2 and P65”.
- Insert the relevant citation to the sentence in line 336-339.
- Move the conclusion part (line 457-461) to the last paragraph of the discussion section.
- Please revise the 4.5 section Cell proliferation and Colony formation assay (line 405-410). Some sentences are confusing. For example, “Using the MTT assay. For colony formation assay, about 1000 cells were seeded… For Colony formation, 5000 cells were seeded…”
- Rewrite the sentence in line 412-414. It is too long and unclear.
- Please read the manuscript carefully and correct typos and spelling errors. Here are some examples:
“PAPR1” in line 16, 20, 67, 259, 279 & 376 should be “PARP1”.
“regulate transcription CCL2” in line 24 should be “regulate transcription of CCL2”.
“upon CCL2. Treatment.” in line 27 should be “upon CCL2 treatment.”
“+-“ in line 160 should be “±”.
“Experiment” in line 197 should be “experiment”.
“, to determine this” in line 230 should be “To determine this”.
Author Response
Reviewer-3 Comments:
Comments and Suggestions for Authors:
This is an interesting report. It shows that PARP1 regulates CCL2 transcription through interaction with NFκB P65 subunit. In turn, CCL2 can also positively modulate PARP1 activity. The crosstalk between PARP1 and CCL2 regulates invasiveness in breast cancer.
Response: We are thankful to the reviewer of the comments. We have addressed all concerns in the revised manuscripts with changes in the text, figures with updated references as follows.
The following issues need to be addressed to clarify and strengthen the manuscript:
- Other cytokines, such as CCL3, are also down-regulated under both PARP inhibition and knock down, which means PARP-1 could regulate the transcription of other cytokines. Please explain why you only focus on CCL-2 regulation?
Response: We chose to follow up with CCL2 due to its known role cancer invasion, metastasis and our previous experience working with CCL2 in tumor cells. However, this is a good suggestions for our future work on cytokines and tumor biology. We would like to follow up with other cytokines in future to understand the role of PARP1 in tumor invasion.
- In the introduction section, there are no relevant citations describing PJ34. Please provide.
Response: We have included the following citation “Abdelkarim, G.E., Gertz, K., Harms, C., Katchanov, J., Dirnagl, U., Szabo, C., & Endres, M. (2001). Protective effects of PJ34, a novel, potent inhibitor of poly(ADP-ribose) polymerase (PARP) in in vitro and in vivo models of stroke. International Journal of Molecular Medicine, 7, 255-260. https://doi.org/10.3892/ijmm.7.3.255”. In the result section where Pj34 is mentioned for the first time.
- In Fig. 1A, please specify which cell lines are triple negative cell lines.
Response: In our response above to Reviewer 2 we mentioned the change. “We have indicated that in the figure 1A showing the TNBC lines on the left and ER/PR+ lines on the right side of the blot.”
- Please double check the statistical analysis used in Fig. 2, 3 & 4. In these figures, there are 1 control group and 2-3 independent treatment groups. Running two t-tests on the same data will increase type 1 error. Therefore, one-way ANOVA followed by a post-hoc analysis should be used.
Response: The previous review also raised this concern. We are addressing the issue as follows:” For statistical analysis, we understand that it might not be correct to assume that we have equal variance or normal distribution for the Cq values in RTPCR. Therefore nonparametric tests or ANOVA will be better. We carried out ANOVA for the RTPCR results and compared to the untreated samples.”
- In line 334 and 335, “transcription of CCL2 and CCL2” should be “transcription of CCL2 and P65”
Response: We have broken up the sentences for clarity. “we show that PARP1 interacts with the NFκB P65 subunit, thereby maintaining transcription of CCL2. CCL2, in turn, can positively modulate PARP1 activity in cancer cells.”
- Insert the relevant citation to the sentence in line 336-339.
Response: We have included the following citation “Ohanna M, Giuliano S, Bonet C, et al. Senescent cells develop a PARP-1 and nuclear factor-{kappa}B-associated secretome (PNAS). Genes Dev. 2011;25(12):1245‐1261. doi:10.1101/gad.625811”
- Move the conclusion part (line 457-461) to the last paragraph of the discussion section.
Response: We included the conclusion section as required by the journal separately from the discussion. The conclusion section was part of template provided by MDPI Cancers journal website.
- Please revise the 4.5 section Cell proliferation and Colony formation assay (line 405-410). Some sentences are confusing. For example, “Using the MTT assay. For colony formation assay, about 1000 cells were seeded… For Colony formation, 5000 cells were seeded…”
Response: We have corrected the cell numbers to 100cells/well. We have stated in the materials methods section as it as “Cells were plated in each well of 96-well plates and assessed for growth inhibition after 48-72 hrs using the MTT assay. For colony formation assay, about 1000 cells/well were seeded into low attachment 48 well plates for 7days in triplicates with or without PJ34 treatment. Bright-field pictures were taken with a Leica inverted microscope at 10x magnification.”
- Rewrite the sentence in line 412-414. It is too long and unclear.
Response: We have corrected the as in the previous comment. We also changed the wound healing method as “Wound healing assay was performed on 96-well plate. The wound was created using Biotek (VT, USA) autoscratcher. The wound area was imaged using Gen5 software on Cytation 1 imager for every hour for 16 hours after treatment. Wound closing was measured using Gen 5 software and graph created using Microsoft Excel.”
- Please read the manuscript carefully and correct typos and spelling errors. Here are some examples:
“PAPR1” in line 16, 20, 67, 259, 279 & 376 should be “PARP1”.
“regulate transcription CCL2” in line 24 should be “regulate transcription of CCL2”.
“upon CCL2. Treatment.” in line 27 should be “upon CCL2 treatment.”
“+-“ in line 160 should be “±”.
“Experiment” in line 197 should be “experiment”.
“, to determine this” in line 230 should be “To determine this”.
Response: We have corrected the errors in the text. We have corrected “PAPR1” to “PARP1”. We have removed the “period” after CCL2. We have changed “+-“ in Figure legend 2 to “±”. We have corrected “Experiments” in figure legend 3 to “experiments”. We have changed “comma” to “period” and changed “, to determine this” “. To determine this,”.

Round 2
Reviewer 1 Report
The authors have successfully addressed all major and minor concerns. A few more suggestions on how to further improve their manuscript is below.
1) Please replace "PARP1PARP1" with "PARP1" (line 67).
2) Please remove "that of" (line 84).
3) Please change "Bottom left" to "bottom right" (line 103).
4) Please move the sentence "Bottom left showing number of differentially expressed genes that belong to the molecular pathway category" (line 103) to the end of the figure legend in order to maintain consistent read order sequence from left to right.
5) Please change "Bottom Right" to "bottom left" (line 104).
6) "PARP1 inhibition" (line 105) should be followed by "(not shown)" to distinguish the absence of MB-231 cell transcriptomics data upon PJ34 treatment.
7) Please add reference to Supplementary figure 1 after the sentence "Along with that, we also used RNA harvested from MB-231 cells with PJ34 at 10 μM doses overnight" (line 108).
8) There seems to be a typographical error in "1059" (line 126) as this number does not correspond to "1095" displayed in Supplementary figure 1A. Please swap "5" with "9".
9) There is another typographical error in "ADMA8" (line 140) as this gene is displayed as "ADAM8" in Supplementary figure 2. Please swap "M" with the second "A".
10) Please remove "is" (line 230).
11) Please remove period in "1 hr." (line 240).
12) Please replace "Quantitate" with "Quantitative" (line 251).
13) Please remove "under" (line 255).
14) Please provide scale bar length description for Figure 5A right (line 294).
15) The end of the sentence "as determined by microarray" (line 324) could read better as "as determined by microarray analysis".
16) The phrase "recruitment at" (line 336) could be changed to "their recruitment to".
17) Please change "xenbrowser.net" to "xenabrowser.net" (line 375 and in the legend to Supplementary figure 6).
18) Please replace "Cell Signaling Technologies" with "Cell Signaling Technology" (line 398).
19) Please place a right bracket after "4335-MC-100" (line 400) and "P4707" (line 503).
20) Please revise the sentence "PJ34 hydrochloride (Sigma, MO, USA, 401 P4365)" (line 401) as it does not have a verb and a period at the end.
21) Please change "2279-MC-050" to "279-MC-050" (line 406).
22) There seems to be an extra "(" in "((MsigDB oncogenic signature)" (line 416) and in "Actual ES/mean(ES" in the legend to Supplementary figure 1B.
23) "minutes" could be shortened to "min" (lines 459, 496, 499, 506).
24) Please add "cells" after "BT549" (line 468).
25) Please change "manufacture’s" to "manufacturer’s" (line 472).
26) Please replace "Puromycine" with "Puromycin" (lines 473 and 475).
27) Please change "genrate" to "generate" (line 474).
28) Please place a period after "membrane" (line 500).
29) Please change "seedd" to "seeded" (line 503).
30) Please change "follwoed" to "followed" (line 506).
31) Please replace "vectshield" to "Vectashield" (line 507).
32) Please remove colon from "Assay:" (line 509).
33) Please replace "crried" with "carried" (line 513).
34) Please replace "Nxt" with "NxT" (line 513).
35) Please change "supplementary" to "Supplementary" (line 537 and in the legend to Supplementary figure 11).
36) Please change "represents" to "represent" in the legend to Supplementary figure 1B.
37) There are two "are" in the sentence "Bottom right are 7AAD positive cells are possibly necrotic/dead" in the legend to Supplementary figure 7.
38) There were no excel files attached as part of the supplementary file appendix.
Author Response
Reviewer 1 comments:
The authors have successfully addressed all major and minor concerns. A few more suggestions on how to further improve their manuscript is below.
Response: We are thankful for the review. We have provided corrections and answers to concerns as below as well as other spelling corrections.
1) Please replace "PARP1PARP1" with "PARP1" (line 67).
Response: We have corrected the error to remove the extra "PARP1".
2) Please remove "that of" (line 84).
Response: We have corrected the error to remove "that of."
3) Please change "Bottom left" to "bottom right" (line 103).
Response: We have corrected the error to (bottom right).
4) Please move the sentence "Bottom left showing the number of differentially expressed genes that belong to the molecular pathway category" (line 103) to the end of the figure legend in order to maintain consistent read order sequence from left to right.
Response: We have moved the line to the end. We have moved the sentence "Genes" column bottom right showing the number of differentially expressed genes that belong to the molecular pathway category." to the end of the legend.
5) Please change "Bottom Right" to "bottom left" (line 104).
Response: We have corrected the error to "bottom left."
6) "PARP1 inhibition" (line 105) should be followed by "(not shown)" to distinguish the absence of MB-231 cell transcriptomics data upon PJ34 treatment.
Response: We have added (not shown) after inhibition.
7) Please add a reference to Supplementary figure 1 after the sentence, "Along with that, we also used RNA harvested from MB-231 cells with PJ34 at 10 μM doses overnight" (line 108).
Response: We have added "(Supplementary Figure 2)." after the sentence. The supplementary figure numbers changes due to moving "Supplementary Figure 7" to "Supplementary Figure 1" as per Reviewer 2's advice.
8) There seems to be a typographical error in "1059" (line 126) as this number does not correspond to "1095" displayed in Supplementary figure 1A. Please swap "5" with "9".
Response: We have corrected to "1095".
9) There is another typographical error in "ADMA8" (line 140) as this gene is displayed as "ADAM8" in Supplementary figure 2. Please swap "M" with the second "A."
Response: We have corrected "ADAM8".
10) Please remove "is" (line 230).
Response: We have removed "is" after p65.
11) Please remove the period in "1 hr." (line 240).
Response: We have removed the period. (Current line 239)
12) Please replace "Quantitate" with "Quantitative" (line 251).
Response: We have corrected to "Quantitative." (Current line 249)
13) Please remove "under" (line 255).
Response: We have removed "under." (Current line 253).
14) Please provide a scale bar length description for Figure 5A right (line 294).
Response: We have added "Scale bar 20 µm" in the legend. (Current line 293).
15) The end of the sentence "as determined by microarray" (line 324) could read better as "as determined by microarray analysis."
Response: We have added "analysis" after microarray in the legend. (Current line 322).
16) The phrase "recruitment at" (line 336) could be changed to "their recruitment to."
Response: We have modified to "PARP1 and P65 followed by their recruitment to the CCL2 promoter." (Current line 334).
17) Please change "xenbrowser.net" to "xenabrowser.net" (line 375 and in the legend to Supplementary figure 6).
Response: We have corrected "xenabrowser.net." (Current line 375) as well as in Supplementary Figure 7.
18) Please replace "Cell Signaling Technologies" with "Cell Signaling Technology" (line 398).
Response: We have corrected to "Technology."
19) Please place a right bracket after "4335-MC-100" (line 400) and "P4707" (line 503).
Response: We have corrected these mistakes. (Current lines 400 and 502).
20) Please revise the sentence "PJ34 hydrochloride (Sigma, MO, USA, 401 P4365)" (line 401) as it does not have a verb and a period at the end.
Response: We have changed it to "PJ34 hydrochloride was purchased from Millipore Sigma". (Current line 401-402).
21) Please change "2279-MC-050" to "279-MC-050" (line 406).
Response: We have corrected to 279-MC-050. (Current line 406)
22) There seems to be an extra "(" in "((MsigDB oncogenic signature)" (line 416) and in "Actual ES/mean (ES" in the legend to Supplementary figure 1B.
Response: We have removed the extra parenthesis. We have corrected the supplementary figure 2 to "[Actual ES/mean (ES against all permutation of the dataset)]"
23) "minutes" could be shortened to "min" (lines 459, 496, 499, 506).
Response: We have changed to "min." (Current line 458, 495, 498 and 505).
24) Please add "cells" after "BT549" (line 468).
Response: We have added "cells" after BT549. (Current line 467).
25) Please change "manufacture's" to" manufacturers" (line 472).
Response: We have corrected the mistakes. (Current line 471).
26) Please replace "Puromycine" with "Puromycin" (lines 473 and 475).
Response: We have corrected the mistakes. (Current line 472 and 474).
27) Please change "genrate" to "generate" (line 474).
Response: We have corrected the mistakes. (Current line 473).
28) Please place a period after "membrane" (line 500).
Response: We have added the period. (Current line 499).
29) Please change "seedd" to "seeded" (line 503).
Response: We have corrected to "seeded." (Current line 502).
30) Please change "follwoed" to "followed" (line 506).
Response: We have corrected "followed." (Current line 505)
31) Please replace "vectshield" to "Vectashield" (line 507).
Response: We have corrected "Vectashield." (Current line 506).
32) Please remove the colon from "Assay:" (line 509).
Response: We have removed the colon from "4.12: Annexin V Apoptosis Assay". 33) Please replace "crried" with "carried" (line 513).
Response: We have corrected to "carried." (Current line 512)
34) Please replace "Nxt" with "NxT" (line 513).
Response: We have corrected "NxT." (Current line 512)
35) Please change "supplementary" to "Supplementary" (line 537 and in the legend to Supplementary figure 11).
Response: Since we reordered Supplementary figures with figure 7 as figure 1 in the Supplementary pdf, we have modified the legend for Supplemental materials accordingly. (Current lines 528-538).
36) Please change "represents" to "represent" in the legend to Supplementary figure 1B.
Response: We have corrected to remove the "s" to read. "The lines also represent leading-edge subsets." In the legend of now Supplementary figure 2B.
37) There are two "are" in the sentence "Bottom right are 7AAD positive cells are possibly necrotic/dead" in the legend to Supplementary figure 7.
Response: We have modified it to "Bottom left quadrant: 7AAD negative are live cells. Bottom right quadrant: 7AAD positive cells are possibly necrotic/dead." We have reordered the figures as per Reviewer 2's advice. This is currently Supplementary figure 1.
38) There were no excel files attached as part of the supplementary file appendix.
Response: We have added the excel data in the supplementary pdf file since the option to upload excel was not available. We have removed the reference to the excel file in the description of the Supplementary materials in the main text and mentioned it as "Supplementary tables with gene lists from the whole transcriptome analysis."

Reviewer 2 Report
The author have answered my concerns and edited the manuscripts.
The supplemental file could be relabeled it is strange to begin with suppl 7.
line 446 8µM should read 8µm
be consistent with tabulation in each paragraph
Author Response
Reviewer 2 Comments:
The author has answered my concerns and edited the manuscripts.
Response: We are thankful for the review. We have provided corrections to the points, as stated below.
- The supplemental file could be relabeled; it is strange, to begin with, suppl 7.
Response: We have reordered the supplementary figure numbers and made changes to the text. We have created this figure as "Supplementary Figure 1". (Current line 79).
- line 446 8µM should read 8µm
Response: We have corrected this to show the proper unit. (Current line 445).
- be consistent with tabulation in each paragraph
Response: We have made efforts to be consistent. We modified the indent and spacing to make paragraphs follow similar formatting throughout the text. We have made all figure references bold in the text for consistency as well as the same capitalization for figure legend reference in the main text.
